# Mitigating Embedding Collapse in Diffusion Models for Categorical Data

## Abstract

Latent diffusion models have enabled continuous-state diffusion models to handle a variety of datasets, including categorical data. However, most methods rely on fixed pretrained embeddings, limiting the benefits of joint training with the diffusion model. While jointly learning the embedding (via reconstruction loss) and the latent diffusion model (via score matching loss) could enhance performance, our analysis shows that end-to-end training risks embedding collapse, degrading generation quality. To address this issue, we introduce CATDM, a continuous diffusion framework within the embedding space that stabilizes training. We propose a novel objective combining the joint embedding-diffusion variational lower bound with a Consistency-Matching (CM) regularizer, alongside a shifted cosine noise schedule and random dropping strategy. The CM regularizer ensures the recovery of the true data distribution. Experiments on benchmarks show that CATDM mitigates embedding collapse, yielding superior results on FFHQ, LSUN Churches, and LSUN Bedrooms. In particular, CATDM achieves an FID of 6.81 on ImageNet $256 \times 256$ with 50 steps. It outperforms non-autoregressive models in machine translation and is on a par with previous methods in text generation.

## 1 Introduction

Continuous-state diffusion models (CSDMs) (Sohl-Dickstein et al., 2015; Ho et al., 2020; Song et al., 2020b) have recently achieved notable success in various application domains, including computer vision (He et al., 2024; Dhariwal & Nichol, 2021; Chen et al., 2022), natural language processing (Li et al., 2022), and audio (Chen et al., 2021; Kong et al., 2021; Hernandez-Olivan et al., 2023). These probabilistic models learn the inverse of a Markov chain that gradually converts data into pure Gaussian noise, using noise-conditioned score functions (i.e., gradients of log density), which are defined only for continuous data. The core concept is to progressively recover the original data distribution using a learned transition kernel. Diffusion models are notable for their high-fidelity generation (Dhariwal & Nichol, 2021; Lai et al., 2023a;b). They offer stable and relatively efficient training procedures that contribute to their success. Recent advances, such as consistency models (Song et al., 2023; Kim et al., 2023; Luo et al., 2023), have further enhanced diffusion models by reducing the number of sampling steps, making them more practical for real-world applications.

Despite the widespread popularity of CSDMs, their extension to categorical data remains limited. Previous attempts to address this limitation (Austin et al., 2021; Hoogeboom et al., 2021b; Campbell et al., 2022; Sun et al., 2023; Lou et al., 2023) have focused on discrete-state diffusion models (DSDMs), which define discrete corruption processes for categorical data and mimic Gaussian kernels used in continuous space. For instance, D3PMs (Austin et al., 2021) implemented the corruption process as random masking or token swapping and learned to reverse this process from the noisy data. However, unlike continuous diffusion processes, these corruption techniques do not gradually erase the semantic meaning of the data, which ideally would place similar tokens close together and dissimilar ones further apart. This discrepancy leads to an unsmooth reverse procedure and limits their ability to fully exploit the advancements made in CSDMs.

Alternatively, categorical data can be mapped into a continuous embedding space (Vahdat et al., 2021; Rombach et al., 2022; Sinha et al., 2021), followed by the application of CSDMs with Gaussian kernels, which enables progressive learning signals (Ho et al., 2020) and fine-grained sampling. This approach has been successful in various domains. However, it may not inherently yield comparable

results (Li et al., 2022; Strudel et al., 2022; Dieleman et al., 2022). First, it requires a well-trained embedding for each new dataset (Li et al., 2022) before training CSDMs. Since the embedding space and the denoising model are not trained end-to-end, this can result in suboptimal performance. Second, jointly training both components is challenging and prone to the *embedding collapse problem* (Dieleman et al., 2022; Gao et al., 2024), where all embeddings converge to a single vector. While this convergence helps the diffusion model predict clean embeddings, it does not result in a meaningful model and instead leads to poor generation. To alleviate embedding collapse, previous work have explored normalizing embedding vectors to a fixed bounded norm (Dieleman et al., 2022) or mapping the predicted embedding to its nearest neighbor within the finite set of vectors (Li et al., 2022). However, our experiments have shown that these manipulations do not yield satisfactory results in practice.

In response, this paper presents a simple but effective method called *Enhanced Embedding for CATegorical Data in Diffusion Models* (CATDM), specifically designed to address the embedding collapse problem. Our key contributions are summarized as follows:

(1) We suggest that embedding collapse is driven by two factors: (1) the reconstruction loss does not provide enough learning feedback to maintain diverse embeddings, leading to collapse, and (2) the denoising score matching disproportionately influences the variational lower-bound objective, overshadowing the contributions of the embeddings in the reconstruction loss.

(2) We introduce several techniques to prevent trivial or collapsed embeddings. A new loss function is proposed to stabilize training. In particular, we enforce a *Consistency Matching (CM)* regularization that requires the model predictions to remain consistent over time. This ensures that the model produces stable outputs throughout the generation process. To enhance generation quality, we implement (i) shifted cosine noise schedule and (ii) random dropping of embeddings.

(3) We theoretically show that our newly proposed CM regularization helps learn the true data density (Theorem 1) at its optimal. Furthermore, we connect this CM regularizer to heuristic regularizations found in the literature (Dieleman et al., 2022; Gao et al., 2024) (Proposition 2).

Comprehensive experiments across a range of benchmark datasets are conducted to evaluate CATDM. This comprehensive evaluation provides an in-depth analysis of the adaptability and performance of our proposed approach in image generation, text generation, and machine translation tasks. The results show that CATDM effectively mitigates the embedding collapse issue and consistently outperforms several baseline methods. Although our main focus is vision and text generation, CATDM can be applied to any task involving categorical variables. CATDM performs on a par with baselines in text generation and machine translation, and achieves FID scores of $7.25$ on FFHQ, $4.99$ on LSUN Churches, $4.16$ on LSUN Bedrooms, and $6.81$ on ImageNet $256 \times 256$ in image generation, outperforming discrete-based models.

## 2 DIFFUSION MODEL IN EMBEDDING SPACE

Consider a sequence of tokens $\mathbf{x} = [x_1, \ldots, x_M]$, where each element belongs to one of the $K$ categories, i.e., $x_i \in \{1, \ldots, K\}$. Given a dataset of observations, the goal of generative models is to estimate the probability mass function $P_{\text{data}}(\mathbf{x})$. To handle discontinuity, we propose using continuous embeddings, where different categories are represented by real-valued vectors in a continuous latent space. Specifically, let $\boldsymbol{\phi} = \{\boldsymbol{e}_1, \ldots, \boldsymbol{e}_K\}$, where $\boldsymbol{e}_k \in \mathbb{R}^D$, be a learnable codebook, the embeddings of $\mathbf{x}$ are then defined as $\text{EMB}_{\boldsymbol{\phi}}(\mathbf{x}) = [\boldsymbol{e}_{x_1}, \ldots, \boldsymbol{e}_{x_M}]$. We define a sequence of increasingly noisy versions of $\text{EMB}_{\boldsymbol{\phi}}(\mathbf{x})$ as $\mathbf{z}_t$, where $t$ ranges from $t = 0$ (least noisy) to $t = 1$ (most noisy). In the following, we review the variational diffusion formulation (Kingma et al., 2021) in latent space.

**Forward process.** For any $t \in [0, 1]$, the conditional distribution of $\mathbf{z}_t$ given $\mathbf{x}$ is modeled as

$$q_{\boldsymbol{\phi}}(\mathbf{z}_t|\mathbf{x}) = \mathcal{N}(\mathbf{z}_t|\alpha_t \text{EMB}_{\boldsymbol{\phi}}(\mathbf{x}), \sigma_t^2 \boldsymbol{I}),$$

where $\alpha_t$ and $\sigma_t$ are non-negative scalar-value functions of $t$, which determine how much noise is added to the embeddings. We consider a variance-preserving process, i.e., $\alpha_t^2 + \sigma_t^2 = 1$. Under this parameterization, the marginal distribution $q_{\boldsymbol{\phi}}(\mathbf{z}_t)$ is a mixture of Gaussian distributions. Due to the Markovian property by construction, the transition probability distributions are given by

$$q(\mathbf{z}_t|\mathbf{z}_s) = \mathcal{N}(\mathbf{z}_t|\alpha_{t|s}\mathbf{z}_s, \sigma_{t|s}^2 \boldsymbol{I}),$$

where $\alpha_{t|s} = \alpha_t/\alpha_s$ and $\sigma^2_{t|s} = \sigma^2_t - \alpha^2_{t|s}\sigma^2_s$. Conditioned on the clean data $\mathbf{x}$, the forward process posterior distribution is derived as

$$q_{\boldsymbol{\phi}}(\mathbf{z}_s|\mathbf{z}_t, \mathbf{x}) = \mathcal{N}(\mathbf{z}_s|\mu_{\boldsymbol{\phi}}(\mathbf{z}_t, \mathbf{x}; s, t), \sigma^2(s, t)\boldsymbol{I}),$$

where $\mu_{\boldsymbol{\phi}}(\mathbf{z}_t, \mathbf{x}; s, t) = (\alpha_{t|s}\sigma^2_s/\sigma^2_t)\mathbf{z}_t + (\alpha_s\sigma^2_{t|s}/\sigma^2_t)\mathrm{EMB}_{\boldsymbol{\phi}}(\mathbf{x})$ and $\sigma^2(s, t) = \sigma^2_{t|s}\sigma^2_s/\sigma^2_t$.

**Reverse process.** We gradually denoise the latent variables toward the data distribution by a Markov process where the timesteps run backward from $t = 1$ to $t = 0$. Let $\boldsymbol{\theta}$ denote the parameters of the denoising model, the conditional probability distribution $p_{\boldsymbol{\phi},\boldsymbol{\theta}}(\mathbf{z}_s|\mathbf{z}_t; s, t)$ for any $0 \leq s \leq t \leq 1$ in the reverse diffusion process is parameterized by a Gaussian. More specifically, it is given by

$$p_{\boldsymbol{\phi},\boldsymbol{\theta}}(\mathbf{z}_s|\mathbf{z}_t; s, t) = \mathcal{N}(\mathbf{z}_s|\hat{\mu}_{\boldsymbol{\phi},\boldsymbol{\theta}}(\mathbf{z}_t; s, t), \sigma^2(s, t)\boldsymbol{I}), \tag{1}$$

where $\hat{\mu}_{\boldsymbol{\phi},\boldsymbol{\theta}}(\mathbf{z}_t; s, t) = (\alpha_{t|s}\sigma^2_s/\sigma^2_t)\mathbf{z}_t + (\alpha_s\sigma^2_{t|s}/\sigma^2_t)\widehat{\mathrm{EMB}}_{\boldsymbol{\phi},\boldsymbol{\theta}}(\mathbf{z}_t; t)$ and $\widehat{\mathrm{EMB}}_{\boldsymbol{\phi},\boldsymbol{\theta}}(\mathbf{z}_t; t)$ denotes the predicted embeddings of $\mathrm{EMB}_{\boldsymbol{\phi}}(\mathbf{x})$ based on its noisy version $\mathbf{z}_t$.

Following previous work (Dieleman et al., 2022; Gulrajani & Hashimoto, 2024), we parameterize $\widehat{\mathrm{EMB}}_{\boldsymbol{\phi},\boldsymbol{\theta}}(\mathbf{z}_t; t)$ as an average over embeddings. The $i$-element of $\widehat{\mathrm{EMB}}_{\boldsymbol{\phi},\boldsymbol{\theta}}(\mathbf{z}_t; t)$ is given by

$$\left[\widehat{\mathrm{EMB}}_{\boldsymbol{\phi},\boldsymbol{\theta}}(\mathbf{z}_t; t)\right]_{i,:} = \sum_{k=1}^{K} P_{\boldsymbol{\theta}}(\tilde{x}_i = k|\mathbf{z}_t; t)\mathbf{e}_k, \text{ where we write } \tilde{\mathbf{x}} = (\tilde{x}_i)_i. \tag{2}$$

To estimate the posterior probability $P_{\boldsymbol{\theta}}(\tilde{\mathbf{x}}|\mathbf{z}_t; t)$, we use a neural network $f_{\boldsymbol{\theta}}(\mathbf{z}_t; t)$ to predict $K$ logits for each token, followed by a softmax nonlinearity, i.e., $P_{\boldsymbol{\theta}}(\tilde{\mathbf{x}}|\mathbf{z}_t; t) = \prod_{i=1}^{M} \mathrm{softmax}([f_{\boldsymbol{\theta}}(\mathbf{z}_t; t)]_{i,:})$.

**Variational lower bound.** Following Kingma et al. (2021), the negative variational lower bound (VLB) for our diffusion model can be derived as

$$-\log P_{\boldsymbol{\phi},\boldsymbol{\theta}}(\mathbf{x}) \leq D_{\mathrm{KL}}(q_{\boldsymbol{\phi}}(\mathbf{z}_1|\mathbf{x})||p(\mathbf{z}_1)) + \mathbb{E}_{\boldsymbol{\epsilon}\sim\mathcal{N}(\mathbf{0},\boldsymbol{I})}[-\log P_{\boldsymbol{\theta}}(\mathbf{x}|\mathbf{z}_0; 0)] + \mathcal{L}_{\infty}(\mathbf{x}; \boldsymbol{\phi}, \boldsymbol{\theta}), \tag{3}$$

where $\mathbf{z}_t = \alpha_t\mathrm{EMB}_{\boldsymbol{\phi}}(\mathbf{x}) + \sigma_t\boldsymbol{\epsilon}$ and the diffusion loss is simplified to

$$\mathcal{L}_{\infty}(\mathbf{x}; \boldsymbol{\phi}, \boldsymbol{\theta}) = -\frac{1}{2}\mathbb{E}_{\boldsymbol{\epsilon}\sim\mathcal{N}(0,\boldsymbol{I}),t\sim\mathcal{U}(0,1)}\left[\mathrm{SNR}(t)'\|\mathrm{EMB}_{\boldsymbol{\phi}}(\mathbf{x}) - \widehat{\mathrm{EMB}}_{\boldsymbol{\phi},\boldsymbol{\theta}}(\mathbf{z}_t; t)\|^2\right]$$

with $\mathrm{SNR}(t) = \alpha^2_t/\sigma^2_t$ the signal-to-noise ratio. Under certain conditions[1], the prior loss is close to zero as $q_{\boldsymbol{\phi}}(\mathbf{z}_1|\mathbf{x}) \approx \mathcal{N}(0, \boldsymbol{I})$. Unlike CSDMs, the reconstruction loss in our case is important since it involves both denoising and embedding parameters. A remarkable result given by Kingma et al. (2021) is that the diffusion loss is invariant to the noise schedule except at $t = 0$ and $t = 1$.

## 3 ANALYSIS OF EMBEDDING COLLAPSE

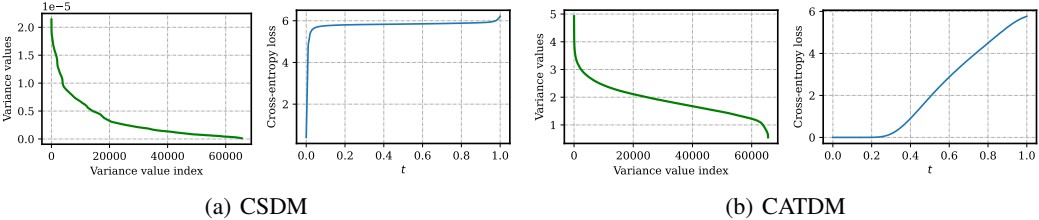

|                | (a) CSDM | (b) CATDM |
| --- | --- | --- |

Figure 1: An illustration of embedding collapse. The variances of the embedding space and cross-entropy loss of (a) CSDM and (b) CATDM.

This section empirically investigates the challenge of jointly learning the embedding and the denoising model. Consider the FFHQ dataset for image generation, where the discrete image tokens are derived from a pretrained VQGAN (Esser et al., 2021). A more detailed description of the experimental setup is described in Section 6.1. Both $\boldsymbol{\phi}$ and $\boldsymbol{\theta}$ are jointly trained by directly minimizing Eq. (3).

---

[1]In theory, we require that $\alpha_1\mathrm{EMB}_{\boldsymbol{\phi}}(\mathbf{x}) = 0$ to ensure that the prior loss is equal zero.

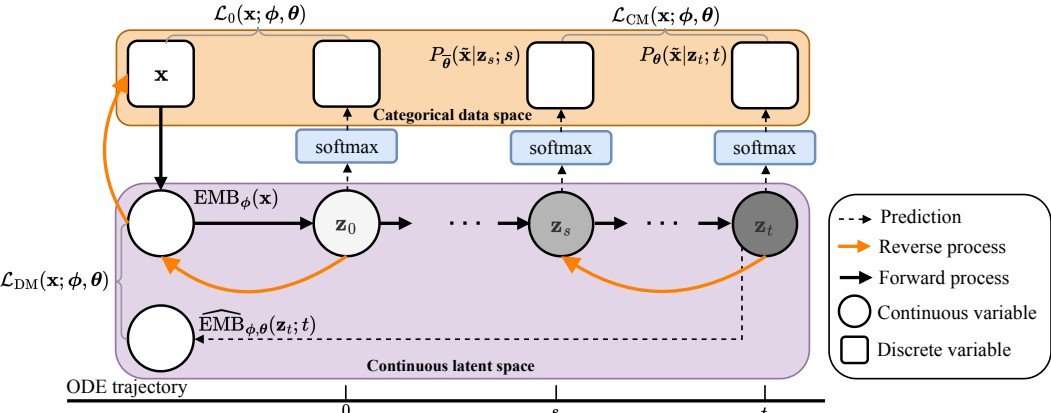

Figure 2: Training procedure of CATDM. Categorical data $\mathbf{x}$ is mapped into continuous embeddings $\text{EMB}_\phi(\mathbf{x})$. We add the consistency-matching loss $\mathcal{L}_{\text{CM}}$ to mitigate emebdding collapse.

This simple model is referred to as CSDM. After training, we then evaluate the dimensionality of the latent space by analyzing the embeddings from 10,000 data examples. Each embedding vector $\text{EMB}_\phi(\mathbf{x})$ has a size of 65,536 in the latent space. We compute the variance of the embeddings along each dimension in sorted order. Figure 1 illustrates the comparison between CSDM and our method CATDM (see Sections 4 and 5). For CSDM, most singular values are zero and the variances are nearly zero. This indicates that the embeddings have collapsed into constant vectors with no variance. In contrast, CATDM produces more diverse embeddings. Note that we use the FFHQ dataset for illustration, but similar results are consistently observed on other datasets such as LSUN Churches, LSUN Bedrooms, and ImageNet. Based on these observations, we suggest the following primary reasons for the embedding collapse.

(i) **The reconstruction loss does not provide enough learning feedback.** Although it penalizes embeddings that are overly similar, this penalization is constrained by the small Gaussian perturbation $\sigma_0$ during the transition from $\text{EMB}_\phi(\mathbf{x})$ to $\mathbf{z}_0$. This is evidenced by the cross-entropy loss being nearly zero only around $t = 0$.

(ii) **The coeficient terms in the diffusion loss encourages constant embeddings**. As $t$ approaches zero, $-0.5\text{SNR}'(t)$ exerts a strong penalization, leading the predictive network to generate constant embeddings as a means to rapidly minimize the diffusion loss. In our ablation studies, we show that a right balance in the objective can help to avoid embedding collapse.

**Why is embedding collapse undesirable?** One of the key strengths of diffusion models is their ability to enable progressive generation (Ho et al., 2020). However, when embeddings become too similar or collapse, this progressive generation is no longer guaranteed. To illustrate this, we use the cross-entropy loss $\mathbb{E}_{\mathbf{x} \sim P_{\text{data}}(\mathbf{x})}[-\log P_\theta(\mathbf{x}|\mathbf{z}_t; t)]$ to measure the uncertainty in distinguishing the true embedding from others. As depicted in Figure 1, the cross-entropy loss for CSDM is low around $t = 0$ but increases rapidly as $t$ arises. The model has a limited time window to generate the global structure of a meaningful embedding, which is required in progressive generation. This indicates that CSDM suffers from an uneven distribution of model capacity. In contrast, for CATDM, the loss gradually increases over time, ensuring that each sampling step equally contributes to resolving uncertainty and facilitating progressive generation. Although the negative VLB values for both models are close (4.83 for CSDM and 4.30 for CATDM), CATDM is preferable due to its ability to support progressive generation.

## 4 CONSISTENCY MATCHING FOR MITIGATING EMBEDDING COLLAPSE

Considering $\text{EMB}_\phi(\mathbf{x})$ as clean data in the continuous space, the evolution of $\text{EMB}_\phi(\mathbf{x})$ over time can be described by the probability flow ordinary differential equation (PF ODE) (Song et al., 2020b). This PF ODE allows a deterministic bijection between the embeddings $\text{EMB}_\phi(\mathbf{x})$ and latent representations $\mathbf{z}_t$. Intuitively, a random noise perturbation $\mathbf{z}_t$ of $\text{EMB}_\phi(\mathbf{x})$ and its relatively nearby

point $\mathbf{z}_s$ along the same trajectory should yield nearly the same prediction. To ensure these consistent outputs for arbitrary $\mathbf{z}_t$, we propose the *Consistency-Matching* (CM) loss

$$\mathcal{L}_{\text{CM}}(\mathbf{x}; \boldsymbol{\phi}, \boldsymbol{\theta}) = \mathbb{E}_{\boldsymbol{\epsilon} \sim \mathcal{N}(0, \boldsymbol{I}), t \sim \mathcal{U}(0,1), s \sim \mathcal{U}(0,t)} \left[ D_{\text{KL}} \big( P_{\overline{\boldsymbol{\theta}}}(\tilde{\mathbf{x}}|\mathbf{z}_s; s) \| P_{\boldsymbol{\theta}}(\tilde{\mathbf{x}}|\mathbf{z}_t; t) \big) \right] , \quad (4)$$

where $\overline{\boldsymbol{\theta}}$ denotes the exponential moving average (EMA) $\overline{\boldsymbol{\theta}} \leftarrow \texttt{stopgrad}(\eta\overline{\boldsymbol{\theta}} + (1-\eta)\boldsymbol{\theta})$ with stop gradient and a rate of $\eta \geq 0$. Here, $\mathbf{z}_t$ is obtained by perturbing $\text{EMB}_{\boldsymbol{\phi}}(\mathbf{x})$ to the noise level $t$, which corresponds to the kernel $q_{\boldsymbol{\phi}}(\mathbf{z}_t|\mathbf{x})$. In our experiment, we simply use the DDIM sampler (Song et al., 2020a) to sample from $\mathbf{z}_t$ to $\mathbf{z}_s$. Under variance preserving settings, it is computed as

$$\mathbf{z}_s = \alpha_s \text{EMB}_{\boldsymbol{\phi}}(\mathbf{x}) + (\sigma_s/\sigma_t)(\mathbf{z}_t - \alpha_t \text{EMB}_{\boldsymbol{\phi}}(\mathbf{x})) .$$

In the following sections, we establish connections between the proposed CM regularizer and existing literature, and demonstrate the theoretical implications of perfectly minimizing the CM objective.

## 4.1 CONNECTION OF CM REGULARIZER TO EXISTING WORKS

When the data distribution $P_{\text{data}}(\mathbf{x})$ is continuous, Eq. (4) recovers the consistency training objective in CSDMs (Song et al., 2023; Kim et al., 2023; Lai et al., 2023b), which matches clean predictions from models along the same sampling PF ODE trajectory. Specifically, for any noisy sample $\mathbf{z}_t$ at time $t$, $P_{\boldsymbol{\theta}}(\tilde{\mathbf{x}}|\mathbf{z}_t; t)$ serves as a deterministic consistency function (Song et al., 2023) $\boldsymbol{h}_{\boldsymbol{\theta}}(\mathbf{z}_t; t)$ predicting the clean sample at time 0 from $\mathbf{z}_t$, regarded as a normal distribution centered around $\boldsymbol{h}_{\boldsymbol{\theta}}(\mathbf{z}_t; t)$ with small variance. Thus, using the closed-form KL divergence of two normal distributions, Eq. (4) becomes:

$$\mathcal{L}_{\text{CM}}(\mathbf{x}; \boldsymbol{\phi}, \boldsymbol{\theta}) \propto \mathbb{E}_{\boldsymbol{\epsilon} \sim \mathcal{N}(0, \boldsymbol{I}), t \sim \mathcal{U}(0,1), s \sim \mathcal{U}(0,t)} \left[ \left\| \boldsymbol{h}_{\overline{\boldsymbol{\theta}}}(\mathbf{z}_s; s) - \boldsymbol{h}_{\boldsymbol{\theta}}(\mathbf{z}_t; t) \right\|_2^2 \right] ,$$

which coincides with objective proposed in (Song et al., 2023; Kim et al., 2023). Here, $\propto$ denotes the omission of multiplicative or additive constants that are independent of the training parameters.

## 4.2 CM REGULARIZER HELPS LEARN TRUE DATA DISTRIBUTION

When the timesteps $t$ are small, the model learns the true categorical distribution through the reconstruction loss. As training progresses, this consistency is propagated to later timesteps, eventually reaching $t = 1$. In other words, the consistency-matching loss encourages the probability distributions of $\mathbf{x}$ in neighboring latent variables to converge. Once the model is fully trained, it consistently produces the same probability distribution for the clean data across the entire trajectory. Since the reconstruction loss enforces the mapping from the embedding space back to categorical data, the learning signal is propagated through the entire ODE trajectory.

Below, we reinforce this intuition by theoretically demonstrating that with the CM regularizer, the true data distribution can be learned, provided that the model prediction $P_{\boldsymbol{\theta}}(\mathbf{x}|\mathbf{z}_t; t)$ can perfectly reconstruct categorical data at $t = 0$ using the following reconstruction loss function:

$$\mathcal{L}_0(\mathbf{x}; \boldsymbol{\phi}, \boldsymbol{\theta}) = \mathbb{E}_{\boldsymbol{\epsilon} \sim \mathcal{N}(0, \boldsymbol{I})} \left[ -\log P_{\boldsymbol{\theta}}(\mathbf{x}|\mathbf{z}_0; 0) \right] .$$

It is important to note that this condition prevents the trivial solution, where $P_{\boldsymbol{\theta}}(\mathbf{x}|\mathbf{z}_t; t)$ remains constant, from occurring during training.

**Theorem 1** (The CM regularizer facilitates learning of the true data density). *Let $(\boldsymbol{\phi}^*, \boldsymbol{\theta}^*)$ be the optimal parameters such that*

$$\mathbb{E}_{\mathbf{x} \sim P_{\text{data}}}[\mathcal{L}_{\text{CM}}(\mathbf{x}; \boldsymbol{\phi}^*, \boldsymbol{\theta}^*)] = 0 \quad \text{and} \quad \mathbb{E}_{\mathbf{x} \sim P_{\text{data}}}[\mathcal{L}_0(\mathbf{x}; \boldsymbol{\phi}^*, \boldsymbol{\theta}^*)] = 0.$$

*Suppose that $\mathbb{E}_{\mathbf{x} \sim P_{\text{data}}} \left[ \|\text{EMB}_{\boldsymbol{\phi}^*}(\mathbf{x})\|^2 \right] < \infty$. Then, it follows that $P_{\boldsymbol{\phi}^*, \boldsymbol{\theta}^*} = P_{\text{data}}$.*

Theorem 1 has an important implication. It indicates that we can accurately learn the true distribution of the data. Although it is less likely to reach the global optimum in practice, we empirically show that consistency-matching loss tends to achieve a solution with fewer sampling steps (see Appendix F).

### 4.3 CM REGULARIZER REDUCES CROSS-ENTROPY OF PREDICTIONS ACROSS ANY TIME

In (Gao et al., 2024; Dieleman et al., 2022), it is suggested that employing the cross-entropy loss:
$$\mathcal{L}_{\mathrm{CE}}(\mathbf{x}; \boldsymbol{\phi}, \boldsymbol{\theta}) = \mathbb{E}_{\mathbf{z}_t, t \sim \mathcal{U}(0,1)} \left[ -\log P_{\boldsymbol{\theta}}(\mathbf{x}|\mathbf{z}_t; t) \right],$$
as a regularizer during the joint training of the embedding and diffusion model can help mitigate embedding collapse, although without a rigorous guarantee. The intuition behind this approach is that there is a discrepancy between the predicted embeddings, which arises from the prediction error of the denoising model. The cross-entropy loss aims to compensate for this discrepancy.

Below, we establish a theoretical connection at the optimal point between our CM regularizer and the cross-entropy regularizer used during training:

**Proposition 2.** *Let* $(\boldsymbol{\phi}^*, \boldsymbol{\theta}^*)$ *be optimal parameters such that:*
$$\mathbb{E}_{\mathbf{x} \sim P_{\mathrm{data}}}[\mathcal{L}_{\mathrm{CM}}(\mathbf{x}; \boldsymbol{\phi}^*, \boldsymbol{\theta}^*)] = 0 \quad and \quad \mathbb{E}_{\mathbf{x} \sim P_{\mathrm{data}}}[\mathcal{L}_0(\mathbf{x}; \boldsymbol{\phi}^*, \boldsymbol{\theta}^*)] = 0.$$
*Then,* $\mathbb{E}_{\mathbf{x} \sim P_{\mathrm{data}}}[\mathcal{L}_{\mathrm{CE}}(\mathbf{x}; \boldsymbol{\phi}^*, \boldsymbol{\theta}^*)] = 0.$

Proposition 2 and Theorem 1 suggest that minimizing the proposed CM regularizer offers a more direct way to learning the true data distribution, avoiding embedding collapse. In practice, training with $\mathcal{L}_{\mathrm{CM}}$ consistently outperforms $\mathcal{L}_{\mathrm{CE}}$ even with few sampling steps (see Table 10 in Appendix).

## 5 ADDITIONAL TECHNIQUES FOR MITIGATING EMBEDDING COLLAPSE

This section introduces several techniques to further mitigate the embedding collapse issue in training the embedding and the denoising model. Comprehensive ablation studies are provided in Tables 1 and 7. An overview of CATDM is given in Figure 2.

### 5.1 WEIGHTING FUNCTION

Although $-\mathrm{SNR}(t)'$ in Eq. (3) provides the correct scaling to treat the VLB as an Evidence Lower Bound, we suspect this weighting function may disrupt the balance between training the reconstruction loss and diffusion loss in practice. Instead of minimizing the diffusion loss, we simplify it as
$$\mathcal{L}_{\mathrm{DM}}(\mathbf{x}; \boldsymbol{\phi}, \boldsymbol{\theta}) = \mathbb{E}_{\boldsymbol{\epsilon} \sim \mathcal{N}(0, \boldsymbol{I}), t \sim \mathcal{U}(0,1)} \left[ \|\mathrm{EMB}_{\boldsymbol{\phi}}(\mathbf{x}) - \widehat{\mathrm{EMB}}_{\boldsymbol{\phi}, \boldsymbol{\theta}}(\mathbf{z}_t; t)\|_2^2 \right].$$
Essentially, we ensure that the loss is evenly distributed over different timesteps. The rationale is that alleviating the error in a large noise level can help the model avoid constant embeddings.

Puting it all together, the overall objective function of CATDM is given by
$$\min_{\boldsymbol{\phi}, \boldsymbol{\theta}} \quad \mathbb{E}_{\mathbf{x} \sim P_{\mathrm{data}}}[\mathcal{L}(\mathbf{x}; \boldsymbol{\phi}, \boldsymbol{\theta})] = \mathbb{E}_{\mathbf{x} \sim P_{\mathrm{data}}}[\mathcal{L}_0(\mathbf{x}; \boldsymbol{\phi}, \boldsymbol{\theta}) + \beta_{\mathrm{DM}}\mathcal{L}_{\mathrm{DM}}(\mathbf{x}; \boldsymbol{\phi}, \boldsymbol{\theta}) + \beta_{\mathrm{CM}}\mathcal{L}_{\mathrm{CM}}(\mathbf{x}; \boldsymbol{\phi}, \boldsymbol{\theta})],$$
where $\beta_{\mathrm{DM}} \geq 0$ and $\beta_{\mathrm{CM}} \geq 0$ are hyperparameters. By tuning $\beta_{\mathrm{DM}}$, we can find the right balance between $\mathcal{L}_0(\mathbf{x}; \boldsymbol{\phi}, \boldsymbol{\theta})$ and $\mathcal{L}_{\mathrm{DM}}(\mathbf{x}; \boldsymbol{\phi}, \boldsymbol{\theta})$.

### 5.2 NOISE SCHEDULE

Determining the right amount of noise added to the embeddings in each timestep can play an important role in both the forward and reverse processes of CATDM. If the embedding norms are large, denoising would be a trivial task for low noise levels. This is not desired because the denoising model has only a small time window to generate the global structure of the meaningful embedding. Instead, we note that adjusting the noise schedule (NS) by shifting its $\log \mathrm{SNR}$ curve (Hoogeboom et al., 2023) is crucial. In particular, we use the shifted cosine noise schedule, defined as
$$\log \mathrm{SNR}(t) = -2 \log \tan(\pi t/2) + s,$$
where $s \in \mathbb{R}$ is the shifting hyperparameter. When $s = 0$, it corresponds to the cosine noise schedule (Nichol & Dhariwal, 2021). Essentially, NS implies different weights in the diffusion loss per noise level (Kingma & Gao, 2023). As illustrated in Figure 3, by moving the curve to the left, it gives more importance for higher degrees of noise. Adjusting NS also increases the speed of noise injection in the forward process when $t \approx 0$, making the generation task more challenging for the model to prevent trivial generation and avoid embedding collapse.

### 5.3 RANDOM DROPPING

Table 1: Results of ablation studies on the FFHQ dataset.

| Method | FID↓ | Prec.↑ | Rec.↑ |
|---|---|---|---|
| CSDM | 77.09 | 0.41 | 0.07 |
| CSDM w $\ell_2$-norm | 52.35 | 0.55 | 0.12 |
| CATDM w/o $\mathcal{L}_{CM}$ | 186.95 | 0.02 | 0.00 |
| CATDM w/o NS | 11.19 | 0.71 | 0.42 |
| CATDM w/o RD | 8.20 | **0.73** | 0.42 |
| CATDM | **7.25** | 0.72 | **0.46** |

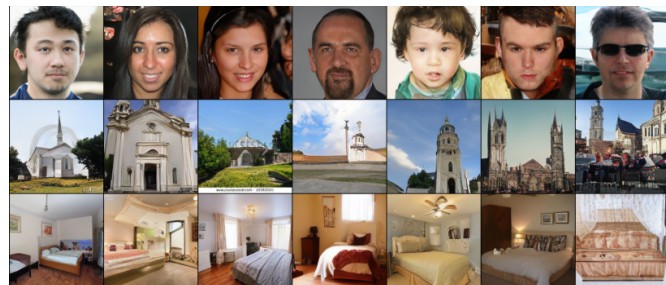

Figure 4: CATDM samples on unconditional image generation.

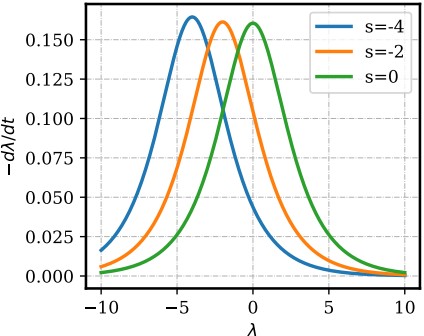

Figure 3: Shifted cosine noise schedule with different shifting factors $s$, $\lambda(t) = \log \text{SNR}(t)$.

Given the noised embeddings, $\mathcal{L}_{CM}$ used in our training objective ensures the same prediction for the posterior probability $P_{\boldsymbol{\theta}}(\tilde{\mathbf{x}}|\mathbf{z}_t; t)$ at any timestep. During joint training, it encourages the model to distinguish the embeddings by increasing their parameter magnitudes. To alleviate this problem, we propose to randomly drop the embeddings. Let $\mathbf{m}_{RD} \in \{0, 1\}^M$ denote a binary mask that indicates which tokens are replaced with a special [mask] token. During training, embeddings of $\mathbf{x}$ become $\text{EMB}_{\phi}(\mathbf{x} \odot \mathbf{m}_{RD})$. Random dropping forces the representations to be more semantic, i.e., similar embeddings should be close to each other (He et al., 2022). This is because only a portion of the embeddings is used to predict the other tokens. It requires the model to understand the relationship between seen and unseen tokens. When similar tokens frequently appear in similar contexts, the model learns to associate these tokens closely in the embedding space, as their contextual meanings are similar.

## 6 EXPERIMENTS

In this section, we assess the performance of CATDM across several benchmark datasets, covering tasks such as image generation, text generation, and machine translation. We also present comprehensive ablation studies to analyze CATDM's performance. Details of the experimental setup are provided in Appendix B.

### 6.1 IMAGE GENERATION

We present experiments covering both conditional and unconditional image generation tasks.

**Datasets.** For unconditional generation tasks, our benchmark consists of three datasets: FFHQ (Karras et al., 2019), LSUN Bedrooms, and LSUN Churches (Yu et al., 2015). The FFHQ dataset contains 70K examples of human faces, while the LSUN Bedrooms dataset contains 3M images of bedrooms, and the LSUN Church dataset contains 126K images of churches. For conditional generation tasks, we use ImageNet (Deng et al., 2009). These datasets are widely used in the literature. All images have a resolution of $256 \times 256$ and VQGAN (Esser et al., 2021) is used to further downsample the images into discrete representations of $16 \times 16$ with a codebook of size 1024.

**Baselines.** We evaluate CATDM against several baselines, including D3PM with uniform transition probabilities (Austin et al., 2021), VQ-Diffusion (Gu et al., 2022), and MaskGIT Chang et al. (2022). Additionally, we include results for CSDM using fixed embeddings (CSDM†), where embeddings are initialized from the pretrained VQGAN codebook and remain fixed throughout training. For evaluation, we report the Fréchet Inception Distance (FID) between 50,000 generated images and real images. We also provide performance metrics in terms of Precision and Recall. For conditional image generation, we use the Inception Score (IS) as an additional metric to measure the image quality.

Table 2: Results for unconditional generation on FFHQ, LSUN Churches, and LSUN Bedrooms. The scores of FID, Precision, and Recall are shown. The **best** and second best results are marked.

| Method | FFHQ | | | LSUN Churches | | | LSUN Bedrooms | | |
|---|---|---|---|---|---|---|---|---|---|
| | FID↓ | Prec.↑ | Rec.↑ | FID↓ | Prec.↑ | Rec.↑ | FID↓ | Prec.↑ | Rec.↑ |
| ***Discrete-Space Diffusion Models*** | | | | | | | | | |
| D3PM Uniform | 9.49 | 0.71 | 0.41 | 6.02 | 0.68 | 0.39 | 6.60 | 0.60 | 0.35 |
| VQ-Diffusion | 8.79 | 0.70 | 0.43 | 6.88 | 0.72 | 0.37 | 7.19 | 0.54 | 0.37 |
| MaskGIT | 11.45 | **0.75** | 0.42 | 5.59 | 0.65 | **0.44** | 8.39 | 0.66 | 0.33 |
| ***Continuous-Space Diffusion Models*** | | | | | | | | | |
| CSDM[†] | 12.66 | 0.73 | 0.38 | 7.88 | **0.76** | 0.36 | 4.93 | 0.71 | 0.38 |
| CATDM (ours) | **7.25** | 0.72 | **0.46** | 4.99 | 0.75 | 0.42 | **4.16** | **0.72** | **0.40** |

**Results.** Table 2 presents the results on unconditional image generation. CATDM consistently achieves the lowest FID scores. Furthermore, we investigate the impact of using pretrained embeddings in CSDM[†] and demonstrate that while it yields satisfactory results, employing trainable embeddings significantly enhances performance. On LSUN Bedrooms, CATDM outperforms the baseline methods by a substantial margin, achieving the highest Precision and Recall scores. These findings underline the superiority of CATDM in generating high-quality samples. The observed improvements in our method compared to discrete diffusion baselines confirm that continuous diffusion models can provide an effective solution for categorical data. Figure 4 illustrates samples generated by CATDM. For additional reference samples generated by CATDM, please refer to Appendix G. Table 3 presents the results for class-conditional image generation tasks. Our method achieves a FID of 6.81 and an IS of 225.31 with 50 sampling steps. CATDM notably outperforms both VQGAN and VQVAE-2 by a substantial margin. Compared to MaskGIT, CATDM provides competitive FID results and exceeds in IS. However, it is important to note, as highlighted by Besnier & Chen (2023), that MaskGIT requires specific sampling adjustments, such as adding Gumbel noise with a linear decay, to improve its FID. In contrast, CATDM operates without such sampling heuristics. In addtiion, CATDM performs better than VQ-Diffusion in both FID and IS metrics. For reference samples generated by CATDM, please refer to Appendix G.

Table 3: Comparison with generative models on ImageNet $256 \times 256$. The results of the existing methods are obtained from their respective published works.

| Model | # params | # steps | FID↓ | IS↑ | Prec.↑ | Rec.↑ |
|---|---|---|---|---|---|---|
| VQGAN (Esser et al., 2021) | 1.4B | 256 | 15.78 | 74.3 | n/a | n/a |
| MaskGIT (Chang et al., 2022) | 227M | 8 | **6.18** | 182.1 | 0.80 | 0.51 |
| VQVAE-2(Razavi et al., 2019) | 13.5B | 5120 | 31.11 | 45.00 | 0.36 | 0.57 |
| BigGAN-deep (Brock et al., 2019) | 160M | 1 | 6.95 | 198.2 | **0.87** | 0.28 |
| Improved DDPM (Nichol & Dhariwal, 2021) | 280M | 250 | 12.26 | n/a | 0.70 | 0.62 |
| VQ-Diffusion (Gu et al., 2022) | 518M | 100 | 11.89 | n/a | n/a | n/a |
| CATDM (ours) | 246M | 50 | 6.81 | **225.31** | 0.84 | 0.38 |

## 6.2 TEXT GENERATION

We evaluate CATDM on unconditional text generation, where the objective is to generate text without any predefined themes or prompts, using a training corpus as the basis for learning.

**Datasets**. We train CATDM on text8 (Mikolov et al., 2014), a character-level language modeling benchmark. The text8 dataset consists of Wikipedia articles with a small vocabulary of 26 letters and a whitespace token. Following (Lou et al., 2023; Austin et al., 2021), we use the same data split and parameterize CATDM on a neural network of similar size.

**Baselines.** CATDM is compared against autoregressive, random-order autoregressive, and other diffusion-based models. Unless otherwise specified, all models are implemented as standard 12-layer transformers. Following (Austin et al., 2021), we report performance using bits per character (BPC).

**Results.** Table 4 shows the results. Autoregressive models, including Discrete Flow (Tran et al., 2019) and Transformer AR (Austin et al., 2021), achieve the best BPC. CATDM outperforms Multinomial

Table 4: Bits per character (BPC) on Text8. (*) Results reported by Shi et al. (2024).

| Method | BPC ↓ |
|---|---|
| ***Random-order Autoregressive Models*** | |
| ARDM (Hoogeboom et al., 2021a) | ≤ 1.43 |
| MAC (Shih et al., 2022) | ≤ 1.40 |
| ***Autoregressive Models*** | |
| IAF/SCF (Ziegler & Rush, 2019) | 1.88 |
| Discrete Flow (Tran et al., 2019) (8 × 3 layers) | **1.23** |
| AR Argmax Flow (Hoogeboom et al., 2021b) | 1.39 |
| Transformer AR (Austin et al., 2021) | **1.23** |
| ***Discrete-State Diffusion Models*** | |
| Mult. Diffusion (Hoogeboom et al., 2021b) | ≤ 1.72 |
| D3PM Uniform (Austin et al., 2021) | ≤ 1.61 |
| D3PM Absorb (Austin et al., 2021) | ≤ 1.45 |
| SEDD Uniform (Lou et al., 2023) | ≤ 1.47 |
| SEDD Absorb (Lou et al., 2023) | ≤ 1.39 |
| MD4 (Shi et al., 2024) | ≤ 1.37 |
| ***Continuous-Space Diffusion Models*** | |
| BFN (Graves et al., 2023) | ≤ 1.41 |
| Plaid (Gulrajani & Hashimoto, 2024) (*) | ≤ 1.48 |
| CATDM (ours) | ≤ 1.54 |

Table 5: Machine translation results. We do not use knowledge distillation. (*) Results obtained by re-running the code.

| Model | # steps | En-De↑ | De-En↑ |
|---|---|---|---|
| ***Autoregressive Models*** | | | |
| Transformer (Vaswani et al., 2017) | n/a | **26.37** | **32.62** |
| ***Discrete-State Diffusion Models*** | | | |
| Mult. Diffusion (Hoogeboom et al., 2021b) | 25 | 3.69 | 20.06 |
| DiffuSeq (Gong et al., 2022) | 1000 | 13.73 | 27.03 |
| SeqDiffuSeq Yuan et al. (2022) | 2000 | 14.37 | 28.65 |
| ***Continuous-Space Diffusion Models*** | | | |
| Diffusion-LM (Li et al., 2022) | 200 | 15.33 | 26.61 |
| CDCD (Dieleman et al., 2022) | 200 | 20.0 | n/a |
| Difformer (Gao et al., 2024) (*) | 20 | 20.89 | 28.30 |
| CATDM (ours) | 20 | 21.67 | 29.61 |

Table 6: FID results for different numbers of sampling steps.

| Steps | 5 | 10 | 15 | 20 | 50 | 100 | 200 |
|---|---|---|---|---|---|---|---|
| Churches | 19.38 | 10.24 | 7.81 | 6.80 | 5.43 | 5.20 | **4.99** |
| Bedrooms | 14.55 | 6.05 | 4.42 | 4.00 | **3.86** | 4.01 | 4.16 |
| FFHQ | 28.80 | 15.55 | 11.44 | 9.57 | 7.56 | 7.34 | **7.25** |
| ImageNet | 12.70 | 7.86 | 7.02 | 6.88 | **6.81** | 6.84 | 6.99 |

Diffusion (Hoogeboom et al., 2021b) and D3PM Uniform (Austin et al., 2021). However, its performance is inferior to that of Plaid (Gulrajani & Hashimoto, 2024) and SEDD (Lou et al., 2023). his discrepancy may be attributed to the fact that CATDM is not specifically designed to maximize the log-likelihood objective. Please refer to Appendix G for reference sentences generated by CATDM.

## 6.3 MACHINE TRANSLATION

Significant efforts have been made to apply non-autoregressive iterative refinement models to the task of machine translation. This section explores the application of CATDM for machine translation.

**Datasets.** We consider two standard datasets, including IWSLT14 German-English (IWSLT14 De-En) (Cettolo et al., 2014) and WMT14 English-German (WMT14 En-De) (Bojar et al., 2014). These datasets are the most popular and widely used benchmarks for evaluating a machine translation system. IWSLT14 De-En consists of transcripts from the TED talks, which are relatively informal, spoken language. WMT14 En-De consists of sentences collected from a variety of sources, including formal as well as less structured text.

**Baselines.** We compare CATDM with the autoregressive Transformer (Vaswani et al., 2017) and other recent diffusion models, including Difformer (Gao et al., 2024), Multinomial (Hoogeboom et al., 2021b), Diffusion-LM (Li et al., 2022), CDCD (Dieleman et al., 2022), DiffuSeq (Gong et al., 2022), and SeqDiffuSeq Yuan et al. (2022). For a fair comparison, we do not adopt any sequence-level knowledge distillation to distill the original training set. For Difformer, we rerun the code provided by the corresponding author. For CATDM, we use the same architecture and setup as that of Difformer, which consists of an encoder-decoder architecture with two distinct Transformer stacks. Unlike autoregresive models where the sequence length is modeled by the EOS token, we explicitly predict the target length using the encoder output. The BLEU score is reported to evaluate our machine translation models.

**Results.** All results, except Difformer and CATDM, were taken from previous studies. As shown in Table 5, CATDM achieves the best results among non-autogressive models, with a BLEU score of 29.61 on on IWSLT14 De-En and 21.67 on WMT14 En-De. Overall, CATDM underperforms compared to the autoregressive model of the same size. This outcome aligns with previous findings (Dieleman et al., 2022), which can be attributed to the fact that non-autoregressive models are not trained to handle repeated or missing tokens.

## 6.4 ABLATION STUDIES

In this section, we conduct various ablation studies. For additional analysis, please see Appendix F.

**Effects of different components.** We investigate the impact of individual components introduced in CATDM on overall performance. The results are presented in Table 1. The baseline method, CSDM, is unable to generate meaningful images. While incorporating an $\ell_2$-norm regularization on the embeddings provides some improvement, it does not completely resolve the collapse issue. CATDM (incorporating our novel components $\mathcal{L}_{\text{CM}} + \text{NS} + \text{RD}$) achieves the best performance. Without RD, the model produces inferior results. Removing NS leads to notable performance degradation. On the other hand, omitting $\mathcal{L}_{\text{CM}}$ results in embedding collapse. These findings highlight the essential role of each component in mitigating the embedding collapse and improving overall performance.

Table 7: Results on different dropping strategies: "linear" means increasing the dropping ratio linearly, "rand_drop" uses a random dropping ratio; and "rand($\gamma$)" uses a fixed dropping ratio of $\gamma$, where $0 \leq \gamma \leq 1$.

|  | FID↓ | Prec.↑ | Rec.↑ |
|---|---|---|---|
| linear | 9.12 | **0.72** | 0.41 |
| rand_drop | 8.44 | 0.71 | 0.42 |
| rand (0.1) | 7.81 | 0.71 | 0.43 |
| rand (0.2) | **7.25** | **0.72** | **0.46** |
| rand (0.3) | 8.45 | 0.70 | 0.43 |
| rand (0.4) | 9.89 | 0.70 | 0.41 |
| rand (0.5) | 9.11 | **0.72** | 0.41 |

**Number of sampling steps.** We analyze the number of steps necessary to obtain high-fidelity samples. Table 6 presents the FID scores corresponding to different numbers of sampling steps. As expected, we observe a decrease in FID as the number of sampling steps increases. However, the improvement becomes marginal after reaching 50 steps. CATDM can accelerate the conventional diffusion models by a large margin, which is a notable advantage compared to ARs.

**Dropping strategies.** We explore various approaches to drop tokens during training. Three distinct strategies are considered. One strategy involves linearly increasing the dropping ratio concerning the time step (**linear**). In this scheme, early steps involve a small portion of tokens being dropped, while in later steps, a higher proportion of tokens are dropped. Another strategy is to randomly select a ratio and drop the tokens according to this ratio (**rand_drop**). Alternatively, a fixed dropping ratio can be employed (**rand(.)**). Table 7 summarizes the results. CATDM performs the best with an appropriately chosen fixed dropping ratio.

## 7 CONCLUSION

This paper introduces CATDM, a continuous diffusion model tailored for modeling categorical distributions, which jointly learns the embeddings and the denoising model. CATDM effectively addresses the issue of embedding collapse. We provide an empirical analysis of this phenomenon, identifying two key mechanisms responsible for collapse: insufficient feedback from the reconstruction loss and the diffusion loss that promotes constant embeddings. Experimental results show that CATDM not only alleviates the embedding collapse problem, but also exceeds the baseline diffusion models.

**Limitations and future work.** In this work, CATDM is implemented using the Transformer architecture, but we emphasize that the architecture choice is orthogonal to the proposed framework and can be extended to other architectures. While we have thoroughly evaluated CATDM on image and text generation tasks, future work will focus on applying it to additional data types, such as graphs and audio. Furthermore, we plan to explore more advanced sampling techniques to improve the overall generation quality of CATDM.

## ETHICS STATEMENT

In conducting AI research focused on developing a diffusion model that addresses the problem of embedding collapse, we are committed to the ethical standards. Our work aims to advance the field of machine learning by improving model robustness. However, we recognize the potential ethical concerns related to the broader societal implications of this work. We acknowledge that AI models can have significant impacts when applied in real-world scenarios. Therefore, we ensure that our research contributes positively to society and does not cause harm.

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

# Appendix

## A RELATED WORK

While autoregressive models (ARs) (Bengio et al., 2000; Brown et al., 2020) appear to dominate categorical data modeling, generating samples from these models incurs significant computational costs. Moreover, controlling these models is challenging because the generation order has to be predetermined (Lou et al., 2023), making them unsuitable for control tasks such as infilling tasks. In contrast, diffusion models predict all tokens simultaneously, allowing efficient and rapid sampling without the sequential attention mechanism of ARs. Below, we present an overview of diffusion models specifically tailored for handling categorical data.

**DSDMs.** The idea is to establish a similar iterative refinement process for categorical data. The corruption process involves transitioning discrete values from one to another. This concept was initially introduced by Sohl-Dickstein et al. (2015) for binary sequence problems. Later, it was extended in multinomial diffusion (Hoogeboom et al., 2021b). Austin et al. (2021) improved discrete diffusion by introducing diverse corruption processes, going beyond uniform transition. Based on the former framework, several extensions have been introduced for image modeling, e.g., MaskGIT (Chang et al., 2022), VQ-Diffusion (Gu et al., 2022), Token-Critic (Lezama et al., 2022), Muse (Chang et al., 2023), and Paella (Rampas et al., 2022). Additionally, Campbell et al. (2022) utilized Continuous Time Markov Chains for discrete diffusion. Despite their initial success, the corruptions introduced by these methods are characterized by their coarse-grained nature, making them inadequate for effectively modeling the semantic correlations between tokens.

**CSDMs.** Li et al. (2022) addressed the challenge of controlling language models (LMs) with Diffusion-LM, a non-autoregressive language model based on continuous diffusion. A similar idea has been introduced in SED (Strudel et al., 2022), DiNoiSer (Ye et al., 2023), CDCD (Dieleman et al., 2022), Bit Diffusion (Chen et al., 2022), Plaid (Gulrajani & Hashimoto, 2024), and Difformer (Gao

et al., 2024). Recently, Meng et al. (2022); Lou et al. (2023) proposed an alternative concrete score function for discrete settings, which captures the surrogate "gradient" information within discrete spaces. However, the challenge of end-to-end training for both embeddings and CSDMs has not been fully addressed in these methods. To avoid embedding collapse, existing techniques either normalize the embeddings (Dieleman et al., 2022) or use heuristic methods (Li et al., 2022), which are not generally effective and may lead to training instability (Dieleman et al., 2022; Strudel et al., 2022).

## B   EXPERIMENTAL SETUP

This section provides architecture details for the denoising model and datasets used in our experiments. Unless specified otherwise, we set the hyperparameters to $\beta_{\text{CM}} = 1$ and $\beta_{\text{DM}} = 0.005$. For embeddings, we use Gaussian initialization $\mathcal{N}(0, D^{-1/2})$. The EMA rate is set to $\eta = 0.99$ and the embedding dimensionality is set to $D = 256$.

### B.1   IMAGE GENERATION

Our prediction network is a bidirectional Transformer (Vaswani et al., 2017). For unconditional generation tasks, the network consists of 15 layers, 8 attention heads, and 512 embedding dimensions (a total of 56M parameters). We apply a dropout rate of 0.1 to the self-attention layers. All models are trained on 4 NVIDIA DGX H100 GPUs with a batch size of 128. The LSUN Bedrooms dataset and FFHQ dataset are trained for 500 epochs, while the LSUN Churches dataset is trained for 100 epochs. For Transformer, we use sinusoidal positional embeddings. To make a fair comparison, all models in Section 6.1 for unconditional image generation are configured with 200 steps for inference.

For conditional image generation on ImageNet, we scale up the model to 24 layers, 16 attention heads, and 768 embedding dimensions (a total of 246M parameters). We train the model for 500 epochs. Following Gu et al. (2022), the conditional class label is injected into the model using Adaptive Layer Normalization (Ba et al., 2016) (AdaLN), i.e., $\text{AdaLN}(\boldsymbol{h}, t) = (1 + \boldsymbol{a}_t)\text{LayerNorm}(\boldsymbol{h}) + \boldsymbol{b}_t$, where $\boldsymbol{h}$ denotes the activation, $\boldsymbol{a}_t$ and $\boldsymbol{b}_t$ are obtained from a linear projection of the class embedding.

### B.2   TEXT GENERATION

We follow the methodologies outlined by Austin et al. (2021) and Lou et al. (2023) for our network architectures and hyperparameters. More spefically, our transformer network consists of 12 layers with 12 heads and a hidden dimension of 768. The model was trained for 500 epochs of batch size 512 and a learning rate of $3 \times 10^{-4}$. For evaluation, the text8 dataset is divided into chunks of length 256 without any preprocessing. Consistent with standard practice, the train/validation/test splits are 90M/5M/5M. The embedding dimensionality is set to 256.

### B.3   MACHINE TRANSLATION

Our model is based on the encoder-decoder Transformer archirtecture (Vaswani et al., 2017). We adopt `fairseq` to implement CATDM. Following (Gao et al., 2024), we use the `transformer-iwslt-de-en` configuration for the IWSLT14 De-En dataset and the `transformer-base` configuration for the WMT14 En-De dataset. The embedding dimension is set to 128. Optimization and learning rate scheduler are the default settings as in (Gao et al., 2024). We explicitly model the target length using maximum log-likelihood.

## C   ALGORITHM PSEUDOCODE

Algorithms 1 and 2 outline the training and sampling procedures of CATDM. For sampling, we discretize time $t \in [0, 1]$ into $N + 1$ points $\{t_n\}_{n=0}^{N}$ such that they satisfy $t_n < t_{n+1}$, $t_0 = 0$, and $t_N = 1$. Starting with Gaussian noise sampled from $\mathbf{z}_{t_N} \sim \mathcal{N}(\mathbf{0}, \mathbf{I})$, we sample $\mathbf{z}_0$ through the ancestral sampling given by $p_{\boldsymbol{\phi},\boldsymbol{\theta}}(\mathbf{z}_{t_{n-1}}|\mathbf{z}_{t_n}; t_{n-1}, t_n)$, which is defined in Eq. (1). Finally, the categorical output $\mathbf{x}$ is obtained from the model $P_{\boldsymbol{\theta}}(\mathbf{x}|\mathbf{z}_0; 0)$. Note that, unlike CSDMs, our model directly outputs the token probabilities for continuous input $\mathbf{z}_{t_n}$ at time step $t_n$.

**Algorithm 1** Training

1: **repeat**
2:     Sample batch of $\mathbf{x} \sim P_{\text{data}}(\mathbf{x})$
3:     $t \sim \mathcal{U}(0, 1);\ s \sim \mathcal{U}(0, t);\ \boldsymbol{\epsilon} \sim \mathcal{N}(\mathbf{0}, \boldsymbol{I});$
       $\mathbf{m}_{\text{RD}} \sim \{0, 1\}^M$
4:     $\mathbf{z}_t := \alpha_t(\text{EMB}_{\boldsymbol{\phi}}(\mathbf{x}) \odot \mathbf{m}_{\text{RD}}) + \sigma_t \boldsymbol{\epsilon}$
5:     Take gradient descent step on
6:         $\nabla_{\boldsymbol{\phi}, \boldsymbol{\theta}} \mathcal{L}(\mathbf{x}; \boldsymbol{\phi}, \boldsymbol{\theta})$
7: **until** converged

**Algorithm 2** Sampling

1: Prepare
       $t_0 := 0 < t_1 < \cdots < t_N := 1$ and
       $\mathbf{z}_{t_N} \sim \mathcal{N}(\mathbf{0}, \boldsymbol{I})$
2: **for** $n = N, N-1, \cdots, 1$ **do**
3:     $\mathbf{z}_{t_{n-1}} \sim p_{\boldsymbol{\phi}, \boldsymbol{\theta}}(\mathbf{z}_{t_{n-1}} | \mathbf{z}_{t_n}; t_{n-1}, t_n)$
4: **end for**
5: $\mathbf{x} \sim P_{\boldsymbol{\theta}}(\mathbf{x} | \mathbf{z}_0; 0)$
6: **return x**

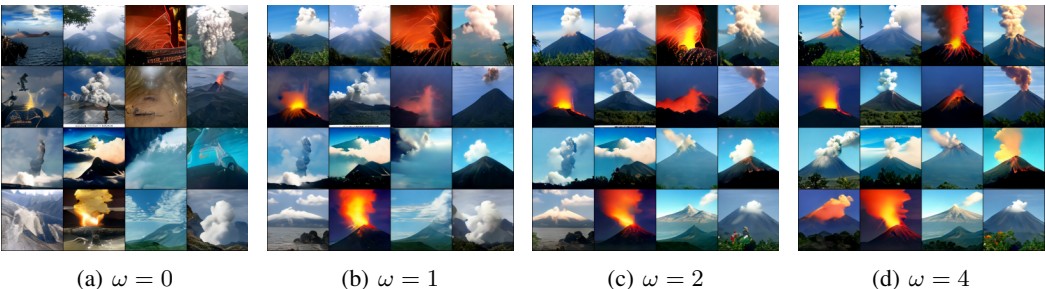

(a) $\omega = 0$      (b) $\omega = 1$      (c) $\omega = 2$      (d) $\omega = 4$

Figure 5: Generated samples of CATDM with $\omega$ ranging from 0 to 4 on ImageNet.

## D    LATENT VARIABLE CLASSIFIER-FREE GUIDANCE

It is important to generate images corresponding to a given condition. In CATDM, the condition is incorporated directly into the prediction network through Adaptive Layer Normalization (Ba et al., 2016). The assumption here is that the network uses both the corrupted input and the condition to reconstruct the input. However, we often observe that CATDM generates outputs that are not correlated well with the condition. The reason is that the corrupted input contains rich information; therefore, the network can ignore the condition during training.

To improve the sample quality of conditional diffusion models, we employ the classifier-free guidance (Ho & Salimans, 2021). Essentially, it guides the sampling trajectories toward higher-density data regions. During training, we randomly drop 10% of the conditions and set the dropped conditions to the null token. During sampling, CATDM predicts the categorical variable $\mathbf{x}$ as follows

$$\log P_{\boldsymbol{\theta}}(\mathbf{x} | \mathbf{z}_t, \mathbf{y}; t) = (1 + \omega) \log P_{\boldsymbol{\theta}}(\mathbf{x} | \mathbf{z}_t, \mathbf{y}; t) - \omega \log P_{\boldsymbol{\theta}}(\mathbf{x} | \mathbf{z}_t; t), \tag{5}$$

where $\omega \geq 0$ denotes the guidance scale and $\mathbf{y}$ denotes the condition. Note that both terms on the right-hand side of Eq. (5) are parameterized by the same model. Figure 5 shows the effects of increasing the classifier-free guidance scale $\omega$.

## E    THEORETICAL ANALYSIS

### E.1    PROOF TO PROPOSITION 2

*Proof.* Since $\overline{\boldsymbol{\theta}^*}$ is a running average of the history of $\boldsymbol{\theta}^*$, it follows that $\overline{\boldsymbol{\theta}^*} = \boldsymbol{\theta}^*$ when $\mathcal{L}_{\text{CM}}(\mathbf{x}; \boldsymbol{\phi}^*, \boldsymbol{\theta}^*) = 0$. Replacing the exponential moving average stop-gradient $\overline{\boldsymbol{\theta}^*}$ in $\mathcal{L}_{\text{CM}}(\mathbf{x}; \boldsymbol{\phi}^*, \boldsymbol{\theta}^*)$ with $\boldsymbol{\theta}^*$, we have

$$\mathcal{L}_{\text{CM}}(\mathbf{x}; \boldsymbol{\theta}^*, \boldsymbol{\phi}^*) = 0 \Leftrightarrow P_{\boldsymbol{\theta}^*}(\tilde{\mathbf{x}} | \mathbf{z}_t; t) = P_{\boldsymbol{\theta}^*}(\tilde{\mathbf{x}} | \mathbf{z}_s; s), \quad \text{for all } \mathbf{z}_t \text{ and } t > s.$$

On the other hand, $\mathcal{L}_0(\mathbf{x}; \boldsymbol{\phi}^*, \boldsymbol{\theta}^*) = 0$ implies

$$\text{one\_hot}(\mathbf{x}) = P_{\boldsymbol{\theta}^*}(\tilde{\mathbf{x}} | \mathbf{z}_0; 0), \tag{6}$$

where $\text{one\_hot}(\mathbf{x})$ contains one-hot encoded representations for each element of $\mathbf{x}$. With $s = 0$, we obtain

$$P_{\boldsymbol{\theta}^*}(\tilde{\mathbf{x}} | \mathbf{z}_t; t) = P_{\boldsymbol{\theta}^*}(\tilde{\mathbf{x}} | \mathbf{z}_0; 0). \tag{7}$$

With Eqs. (6) and (7), for any $\mathbf{x}$ we have

$$P_{\boldsymbol{\theta}^*}(\tilde{\mathbf{x}}|\mathbf{z}_t; t) = \text{one\_hot}(\mathbf{x}) \quad \text{for all } t.$$

This exactly implies $\mathbb{E}_{\mathbf{x} \sim P_{\text{data}}}[\mathcal{L}_{\text{CE}}(\mathbf{x}; \boldsymbol{\phi}^*, \boldsymbol{\theta}^*)] = 0$. $\qquad \square$

### E.2 PROOF TO THEOREM 1

*Proof.* Using the result from Proposition 2, $\mathbb{E}_{\mathbf{x} \sim P_{\text{data}}} \mathcal{L}_{\text{CE}}(\mathbf{x}; \boldsymbol{\phi}^*, \boldsymbol{\theta}^*) = 0$, it directly implies that the denoising loss

$$\mathbb{E}_{\mathbf{x} \sim P_{\text{data}}} \left[ \|\text{EMB}_{\boldsymbol{\phi}^*}(\mathbf{x}) - \widehat{\text{EMB}}_{\boldsymbol{\phi}^*, \boldsymbol{\theta}^*}(\mathbf{z}_t; t)\|_2^2 \right] = 0,$$

as the prediction $\widehat{\text{EMB}}_{\boldsymbol{\phi}^*, \boldsymbol{\theta}^*}(\mathbf{z}_t; t)$ is an average over emebddings, defined in Eq. (2).

Therefore, the following VLB

$$-\mathbb{E}_{\mathbf{x} \sim P_{\text{data}}} \left[ \log P_{\boldsymbol{\phi}^*, \boldsymbol{\theta}^*}(\mathbf{x}) \right] \leq \mathbb{E}_{\mathbf{x} \sim P_{\text{data}}} \left[ D_{\text{KL}}(q_{\boldsymbol{\phi}^*}(\mathbf{z}_1|\mathbf{x}) \| p(\mathbf{z}_1)) \right]$$
$$+ \mathbb{E}_{\mathbf{x} \sim P_{\text{data}}} \left[ \mathcal{L}_0(\mathbf{x}; \boldsymbol{\phi}^*, \boldsymbol{\theta}^*) \right] + \mathbb{E}_{\mathbf{x} \sim P_{\text{data}}} \left[ \mathcal{L}_\infty(\mathbf{x}; \boldsymbol{\phi}^*, \boldsymbol{\theta}^*) \right]$$

implies

$$D_{\text{KL}}\left(P_{\text{data}} \| P_{\boldsymbol{\phi}^*, \boldsymbol{\theta}^*}\right) \leq \mathbb{E}_{\mathbf{x} \sim P_{\text{data}}} \left[ D_{\text{KL}}(q_{\boldsymbol{\phi}^*}(\mathbf{z}_1|\mathbf{x}) \| p(\mathbf{z}_1)) \right].$$

This holds since cross-entropy, entropy, and KL-divergence are non-negative for discrete distributions. We recall that

$$q_{\boldsymbol{\phi}}(\mathbf{z}_t|\mathbf{x}) = \mathcal{N}(\mathbf{z}_t|\alpha_t \text{EMB}_{\boldsymbol{\phi}}(\mathbf{x}), \sigma_t^2 \boldsymbol{I}) \quad \text{and} \quad p(\mathbf{z}_1) = \mathcal{N}(\mathbf{z}_1|\mathbf{0}, \boldsymbol{I}).$$

Using the formula for the KL-divergence of two normal distributions, we have

$$\mathbb{E}_{\mathbf{x} \sim P_{\text{data}}} \left[ D_{\text{KL}}(q_{\boldsymbol{\phi}^*}(\mathbf{z}_1|\mathbf{x}) \| p(\mathbf{z}_1)) \right] = \frac{1}{2} \mathbb{E}_{\mathbf{x} \sim P_{\text{data}}} \left[ D(\sigma_1^2 - 1 - \log \sigma_1^2) + \frac{\|\alpha_1 \text{EMB}_{\boldsymbol{\phi}^*}(\mathbf{x})\|^2}{\sigma_1^2} \right]$$
$$= 0,$$

as $\sigma_1 = 1$, $\alpha_1 = 0$, and $\mathbb{E}_{\mathbf{x} \sim P_{\text{data}}} \left[ \|\text{EMB}_{\boldsymbol{\phi}^*}(\mathbf{x})\|^2 \right] < \infty$. Therefore, $P_{\text{data}} = P_{\boldsymbol{\phi}^*, \boldsymbol{\theta}^*}$. $\qquad \square$

### E.3 CONNECTION OF $\mathcal{L}_{\text{CE}}$ TO DIFFUSION MODEL OBJECTIVE

We show that minimizing $\mathcal{L}_{\text{CE}}$ implicitly regularizes KL divergence of transition density in the embedding space.

**Proposition 3.** *Let $\mathbf{x} \in \mathcal{X}$ and $\mathbf{z}_0 = \text{EMB}_{\boldsymbol{\phi}}(\mathbf{x})$. For any $t \in [0, 1]$ and $\mathbf{z}_t \sim q_{\boldsymbol{\phi}}(\mathbf{z}_t|\mathbf{x})$, we assume that $\log p_{\boldsymbol{\theta}}(\mathbf{z}|\mathbf{z}_t; t)$, as a function of $\mathbf{z}$, is smooth and decays rapidly enough as $\|\mathbf{z}\| \to \infty$. For the discretization of timesteps $1 = t_N > \cdots > t_n \cdots > t_0 = 0$, we have*

$$D_{\text{KL}}\left(q_{\boldsymbol{\phi}}(\mathbf{z}_{t_{n-1}}|\mathbf{z}_{t_n}, \mathbf{z}_0) \| p_{\boldsymbol{\theta}}(\mathbf{z}_{t_{n-1}}|\mathbf{z}_{t_n}; t_{n-1}, t_n)\right) \leq -\log p_{\boldsymbol{\theta}}(\mathbf{x}|\mathbf{z}_{t_n}; t_n).$$

*Proof.* We first prove the "data-processing-type inequality"

$$D_{\text{KL}}\left(\mathbb{E}_{\mathbf{y} \sim Q(\mathbf{y})}[Q(\mathbf{z}|\mathbf{y})] \| \mathbb{E}_{\mathbf{y} \sim P(\mathbf{y})}[Q(\mathbf{z}|\mathbf{y})]\right) \leq D_{\text{KL}}\left(Q(\mathbf{z}) \| P(\mathbf{z})\right). \tag{8}$$

Assume that $P(\mathbf{y}) \neq 0$ and $Q(\mathbf{z}|\mathbf{y}) \neq 0$ almost everywhere for $\mathbf{y}$ and $\mathbf{z}$.

$$D_{\mathrm{KL}}\big(\mathbb{E}_{\mathbf{y}\sim Q(\mathbf{y})}[Q(\mathbf{z}|\mathbf{y})]\|\mathbb{E}_{\mathbf{y}\sim P(\mathbf{y})}[Q(\mathbf{z}|\mathbf{y})]\big)$$

$$= \int \Big(\int Q(\mathbf{z}|\mathbf{y})Q(\mathbf{y})\,\mathrm{d}\mathbf{y}\Big) \cdot \log \frac{\int Q(\mathbf{z}|\mathbf{y})Q(\mathbf{y})\,\mathrm{d}\mathbf{y}}{\int Q(\mathbf{z}|\mathbf{y})P(\mathbf{y})\,\mathrm{d}\mathbf{y}}\,\mathrm{d}\mathbf{z}$$

$$= \int \Big(\int Q(\mathbf{z}|\mathbf{y})P(\mathbf{y})\,\mathrm{d}\mathbf{y}\Big) \cdot \frac{\int Q(\mathbf{z}|\mathbf{y})Q(\mathbf{y})\,\mathrm{d}\mathbf{y}}{\int Q(\mathbf{z}|\mathbf{y})P(\mathbf{y})\,\mathrm{d}\mathbf{y}} \cdot \log \frac{\int Q(\mathbf{z}|\mathbf{y})Q(\mathbf{y})\,\mathrm{d}\mathbf{y}}{\int Q(\mathbf{z}|\mathbf{y})P(\mathbf{y})\,\mathrm{d}\mathbf{y}}\,\mathrm{d}\mathbf{z}$$

$$= \int \Big(\int Q(\mathbf{z}|\mathbf{y})P(\mathbf{y})\,\mathrm{d}\mathbf{y}\Big) \cdot \Psi\Big(\frac{\int Q(\mathbf{z}|\mathbf{y})Q(\mathbf{y})\,\mathrm{d}\mathbf{y}}{\int Q(\mathbf{z}|\mathbf{y})P(\mathbf{y})\,\mathrm{d}\mathbf{y}}\Big)\,\mathrm{d}\mathbf{z}$$

$$\leq \int \int Q(\mathbf{z}|\mathbf{y})P(\mathbf{y}) \cdot \frac{Q(\mathbf{z}|\mathbf{y})Q(\mathbf{y})}{Q(\mathbf{z}|\mathbf{y})P(\mathbf{y})} \cdot \Psi\Big(\frac{Q(\mathbf{z}|\mathbf{y})Q(\mathbf{y})}{Q(\mathbf{z}|\mathbf{y})P(\mathbf{y})}\Big)\,\mathrm{d}\mathbf{z}\,\mathrm{d}\mathbf{y}$$

$$= \int \int Q(\mathbf{z}|\mathbf{y})Q(\mathbf{y}) \log \frac{Q(\mathbf{z}|\mathbf{y})Q(\mathbf{y})}{Q(\mathbf{z}|\mathbf{y})P(\mathbf{y})}\,\mathrm{d}\mathbf{y}\,\mathrm{d}\mathbf{z}$$

$$= \int \Big(\int Q(\mathbf{z}|\mathbf{y})\,\mathrm{d}\mathbf{z}\Big)Q(\mathbf{y}) \log \frac{Q(\mathbf{y})}{P(\mathbf{y})}\,\mathrm{d}\mathbf{y}$$

$$= D_{\mathrm{KL}}\big(Q(\mathbf{y})\|P(\mathbf{y})\big),$$

where the inequality follows from applying Jensen's inequality to the function $\Psi(x) := x \log x$. Now, we let $p_\sigma(\hat{\mathbf{z}}_0|\mathbf{z}_0) := \mathcal{N}\big(\hat{\mathbf{z}}_0|\mathbf{z}_0, \sigma^2 \mathbf{I}\big)$ and $q_{\phi,\sigma}(\mathbf{z}_{t_{n-1}}|\mathbf{z}_{t_n}, \mathbf{z}_0) := \int q(\mathbf{z}_{t_{n-1}}|\mathbf{z}_{t_n}, \hat{\mathbf{z}}_0)p_\sigma(\hat{\mathbf{z}}_0|\mathbf{z}_0)\,\mathrm{d}\hat{\mathbf{z}}_0$. Recall in (continuous state) diffusion model that

$$q(\mathbf{z}_{t_{n-1}}|\mathbf{z}_{t_n}) \approx \mathbb{E}_{\hat{\mathbf{z}}_0 \sim p_\theta(\hat{\mathbf{z}}_0|\mathbf{z}_{t_n})}\big[q(\mathbf{z}_{t_{n-1}}|\mathbf{z}_{t_n}, \hat{\mathbf{z}}_0)\big] =: p_\theta(\mathbf{z}_{t_{n-1}}|\mathbf{z}_{t_n}; t_{n-1}, t_n).$$

By applying (8), we have

$$D_{\mathrm{KL}}\big(q_{\phi,\sigma}(\mathbf{z}_{t_{n-1}}|\mathbf{z}_{t_n}, \mathbf{z}_0)\|p_\theta(\mathbf{z}_{t_{n-1}}|\mathbf{z}_{t_n}; t_{n-1}, t_n)\big)$$

$$= D_{\mathrm{KL}}\big(\int q(\mathbf{z}_{t_{n-1}}|\mathbf{z}_{t_n}, \hat{\mathbf{z}}_0)p_\sigma(\hat{\mathbf{z}}_0|\mathbf{z}_0)\,\mathrm{d}\hat{\mathbf{z}}_0\|\int q(\mathbf{z}_{t_{n-1}}|\mathbf{z}_{t_n}, \hat{\mathbf{z}}_0)p_\theta(\hat{\mathbf{z}}_0|\mathbf{z}_{t_n})\,\mathrm{d}\hat{\mathbf{z}}_0\big)$$

$$\leq D_{\mathrm{KL}}\big(p_\sigma(\hat{\mathbf{z}}_0|\mathbf{z}_0)\|p_\theta(\hat{\mathbf{z}}_0|\mathbf{z}_{t_n}; t_n, 0)\big).$$

Therefore, leveraging the lower semi-continuity property of the KL divergence, we obtain

$$D_{\mathrm{KL}}\big(q_\phi(\mathbf{z}_{t_{n-1}}|\mathbf{z}_{t_n}, \mathbf{z}_0)\|p_\theta(\mathbf{z}_{t_{n-1}}|\mathbf{z}_{t_n}; t_{n-1}, t_n)\big) \leq \liminf_{\sigma \to 0} D_{\mathrm{KL}}\big(q_{\phi,\sigma}(\mathbf{z}_{t_{n-1}}|\mathbf{z}_{t_n}, \mathbf{z}_0)\|p_\theta(\mathbf{z}_{t_{n-1}}|\mathbf{z}_{t_n}; t_n, t_{n-1})\big)$$

$$\leq \liminf_{\sigma \to 0} D_{\mathrm{KL}}\big(p_\sigma(\hat{\mathbf{z}}_0|\mathbf{z}_0)\|p_\theta(\hat{\mathbf{z}}_0|\mathbf{z}_{t_n}; t_n, 0)\big)$$

$$= -\int \log p_\theta(\hat{\mathbf{z}}_0|\mathbf{z}_{t_n})\delta_{\mathbf{z}_0}(\mathrm{d}\hat{\mathbf{z}}_0)$$

$$= -\log p_\theta(\mathbf{z}_0|\mathbf{z}_{t_n}; t_n, 0)$$

$$= -\log p_\theta(\mathbf{x}|\mathbf{z}_{t_n}).$$

$\square$

# F ADDITIONAL ABLATION STUDIES

In this section, we provide additional ablation studies to futher validate our motivations of CATDM.

## F.1 PRETRAINED VS. LEARNABLE EMBEDDINGS

We evaluate the embedding vectors obtained by CATDM against those provided by the pretrained VQGAN on the LSUN Churches dataset. Figure 6 presents the magnitudes of these vectors and the distance matrices between embeddings. Interestingly, our method learns a structure that is quite similar to the pretrained embeddings. Learnable embeddings tend to have larger magnitudes.

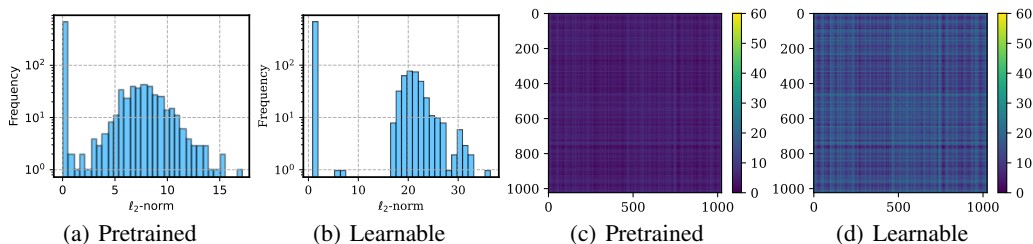

|  (a) Pretrained | (b) Learnable | (c) Pretrained | (d) Learnable |

Figure 6: Visual representation of pretrained and learnable embedding vectors for the LSUN Churches dataset: (a) vector magnitudes for pretrained embeddings, (b) vector magnitudes for learnable embeddings, (c) distance matrix for pretrained embeddings and (d) distance matrix for learnable embeddings. For distance matrices, we compute the Euclidean distances between different embedding vectors.

## F.2 WEIGHTING TERMS

We hypothesize that balancing the reconstruction loss and the diffusion loss is crucial to preventing embedding collapse. In CATDM, this is achieved by tuning the hyperparameter $\beta_{\text{DM}}$. Table 8 presents the FID results on FFHQ for various combinations of $\beta_{\text{CM}}$ and $\beta_{\text{DM}}$. Adjusting these parameters alters the contributions of the diffusion loss and the consistency-matching loss in the objective function. As indicated in the table, when $\beta_{\text{DM}}$ is relatively large, the model still suffers from embedding collapse.

| $\beta_{\text{CM}}$ | $\beta_{\text{DM}}$ | FID ↓ |
|---|---|---|
| 0.01 | 0.01 | 175.46 |
| 0.01 | 1 | 173.28 |
| 1 | 1 | 54.10 |
| 1 | 0.01 | 8.26 |
| 1 | 0.005 | 7.25 |

| $D$ | FID ↓ |
|---|---|
| 64 | 7.90 |
| 128 | 7.20 |
| 256 | 7.25 |
| 768 | 7.42 |
| 1024 | 7.38 |

Table 8: Results on FFHQ with different different hyperparameters $\beta_{\text{CM}}$ and $\beta_{\text{DM}}$.

Table 9: Results on FFHQ when varying the embedding dimensionality.

## F.3 EMBEDDING DIMENSIONALITY

Table 9 shows the influence of embedding dimensionality. We report the FID results on FFHQ when varying the embedding dimensionality. CATDM demonstrates consistent performance across various dimensionalities. As the dimensionality increases, the performance slightly decreases. CATDM achieves the best result when $D = 128$.

## F.4 COMPARISON BETWEEN CROSS-ENTROPY LOSS AND CONSISTENCY-MATCHING LOSS

To demonstrate the effectiveness of our proposal, we conduct experiments by replacing the consistency-matching loss $\mathcal{L}_{\text{CM}}(\mathbf{x}; \boldsymbol{\phi}, \boldsymbol{\theta})$ in CATDM with the cross-entropy loss (CATDM-CE), defined as $\mathcal{L}_{\text{CE}}(\mathbf{x}; \boldsymbol{\phi}, \boldsymbol{\theta}) = \mathbb{E}_{\boldsymbol{\epsilon} \sim \mathcal{N}(0, \boldsymbol{I}), t \sim \mathcal{U}(0,1)}[-\log P_{\boldsymbol{\theta}}(\mathbf{x}|\mathbf{z}_t; t)]$. This cross-entropy loss has been used in several works, including Difformer (Gao et al., 2024) and CDCD (Dieleman et al., 2022), as an additional form of regularization. Table 10 presents the FID results for uncontional image generation when varying the number of sampling steps. Notably, CATDM consistently outperforms the cross-entropy loss regularization, showing significant improvements, particularly when the number of sampling steps is small.

## F.5 ABLATION STUDIES ON IMAGENET

We conduct ablation studies on ImageNet to examine the effects of classifier-free guidance weights and the number of sampling steps. Figure 7(a) shows the FID and IS metrics across various classifier-

Table 10: FID results with different numbers of sampling steps.

| Dataset | Method | Step | | | | | | |
|---|---|---|---|---|---|---|---|---|
| | | 5 | 10 | 15 | 20 | 50 | 100 | 200 |
| Churches | CATDM | **19.38** | **10.24** | **7.81** | **6.80** | **5.43** | **5.20** | **4.99** |
| | CATDM-CE | 21.96 | 11.93 | 8.90 | 7.69 | 5.71 | 5.22 | 5.45 |
| Bedrooms | CATDM | **14.55** | **6.05** | **4.42** | **4.00** | **3.86** | **4.01** | **4.16** |
| | CATDM-CE | 16.69 | 7.89 | 6.60 | 4.81 | 4.10 | 4.13 | 4.50 |
| FFHQ | CATDM | **28.80** | **15.55** | **11.44** | **9.57** | **7.56** | **7.34** | **7.25** |
| | CATDM-CE | 36.57 | 20.45 | 14.68 | 12.14 | 8.60 | 7.79 | 7.60 |

free guidance weight values. Additionally, Figure 7(b) presents the FID and IS results as we vary the number of sampling steps. There is a clear trade-off between fidelity represented by FID and quality represented by IS. CATDM achieves the best FID results when $\omega = 1$.

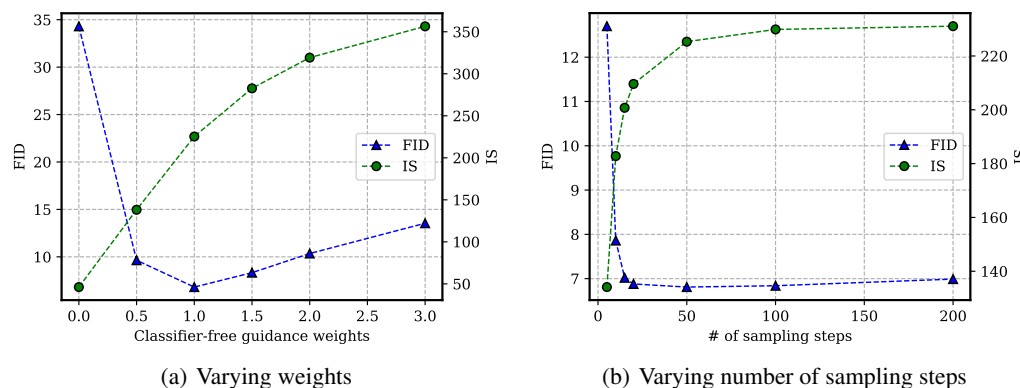

(a) Varying weights

(b) Varying number of sampling steps

Figure 7: Ablation studies for FID vs IS on ImageNet when (a) varying classifier-free guidance weights and (b) varying number of sampling steps.

# G ADDITIONAL SAMPLES

In this section, we present additional samples generated by CATDM. For unconditional image generation, Figures 8, 9, and 10 show the generated samples from CATDM trained on FFHQ, LSUN Churches, and LSUN Bedrooms, respectively. Figure 11 visualizes the conditional samples from ImageNet. All images are at a resolution of $256 \times 256$. Table 11 provides examples of translation, while Table 12 shows samples generated by CATDM trained on the text8 dataset.

Table 11: Translation results of CATDM.

| | **German to English** |
|---|---|
| Source | wir befinden uns also in einer wunderbaren situation mit strom in der reichen welt . |
| Target | and so , we're in a wonderful situation with electricity in the rich world . |
| Generation | so we're ourselves in a wonderful situation with electricity in the rich world . |
| Source | die vereinigten staaten haben heute die höchste inhaftierungsrate der welt . |
| Target | the united states now has the highest rate of incarceration in the world . |
| Generation | the united states today have the highest incerdient in the world today . |
| Source | und problem , das ist nicht nur ein technisches problem , es kann auch ein gesellschaftliches problem sein , es kann auch einfach ein zugangsproblem sein , was wie dinge vereinfachen , also eine beliebige problemstellung , eine frage aufzuwerfen , und wie kann man das anders oder wie kann man das besser machen . |
| Target | and a problem , so not only a technical problem , it can also be a social problem , it can also just be an access problem that simplifying things , so any way of looking at a problem , of posing a question , asking how you could do something differently or better . |
| Generation | and problem , it 's just a technical problem , it can also be a social problem , but it may also be a intellectual problem , which simplibles things like , so , any kind of solving solving to throw a question , and how can you do it differently or how can you do it better ? |

| | **English to German** |
|---|---|
| Source | The stakes are high for the fast @-@ growing economy as the discovery of huge offshore gas reserves and coal deposits in the northwest could bring in more than $ 50bn of investment over the next few next years from companies including Rio Tinto , Vale of Brazil , Eni of Italy and Anadarko of the US . |
| Target | Für die schnell wachsende Wirtschaft steht viel auf dem Spiel , denn die Entdeckung riesiger Gasvorkommen vor der Küste und Kohlelager im Nordwesten könnte in den nächsten Jahren Investitionen von über 50 Milliarden US @-@ Dollar von Firmen wie Rio Tinto , Vale aus Brasilien , Eni aus Italien und Anadarko aus den USA ins Land bringen . |
| Generation | Für die schnell wachsende Wirtschaft steht es hoch auf dem Spiel , da die Entdeckung großer Offshore @-@ Gasreserven und Kohlevorlagen im Nordwesten in den nächsten nächsten Jahren mehr als 50 Milliarden Dollar von Unternehmen wie Rio Tinto , Vale of Brazil , Eni von Italien und Anadarko von den USA in die Investitionen bringen könnte . |
| Source | Chips are available everywhere ! |
| Target | Pommes gibt es überall ! |
| Generation | Chips sind überall verfügbar ! |
| Source | He is always either at the hospital or trying to make money for the organization so he can go on these campaigns . |
| Target | Er ist immer entweder im Krankenhaus oder versucht , Geld für die Organisation aufzubringen , damit er auf diese Kampagnen gehen kann . |
| Generation | Er ist immer im Krankenhaus oder versucht , Geld für die Organisation zu verdienen , damit er auf diese Kampagnen gehen kann . |

Table 12: CATDM samples for unconditional text generation on Text8.

| **Sample # 1** |
|---|
| who had it arrived in at least two zero in the effort the main ridical and heterapeutic defeating agreements in was forced by at a trikonous in two zero zero three a chain of academic remedies epidemics have angered to sudan led al iuan t j judah surah tro |

| **Sample # 2** |
|---|
| policy in australia which had also now been japan one nine three two olean gortowstein usa for leda inoch one nine three six giving anguquin sault blair kogartian linkowaki holidiy first balancer of american supporting to kirgham d one nine six zero one ni |

| **Sample # 3** |
|---|
| star it was assigned to one of these economists believed that there is no girl glenner indicates that incarnation cannot tell with evacuation to the partement of the object are european new and famous spring scroogs hearts and impunity the fertile board mi |

| **Sample # 4** |
|---|
| distinction between hydrogen antr and rans errates of the disease spinally induced aglicozedon distribution of hydrogen and inductors with request signals and tropical mice of the first image anti organic diseases in the second case could more in te also i |

| **Sample # 5** |
|---|
| seven eight their decoration was nine zero one eight in the total of one seven eight zero four zero zero the phantom sajyats civilized man who organized the name shipple a hellu mine and the general fire of their fire all the mongols believe the name remai |

| **Sample # 6** |
|---|
| call for the colmington differences to london national people were able to meet the singular team lorestaray and on february one two zero zero six the fater london kokho london team s arms run by the chief london person who created the average incoded cons |

| **Sample # 7** |
|---|
| pecies by early kinds of biots and some of the human evolution of anti capitalism is constructing invasion and is useful to no anti generated any intelligible free all over the world in which a morphised freedom there is included to represent men s reforme |

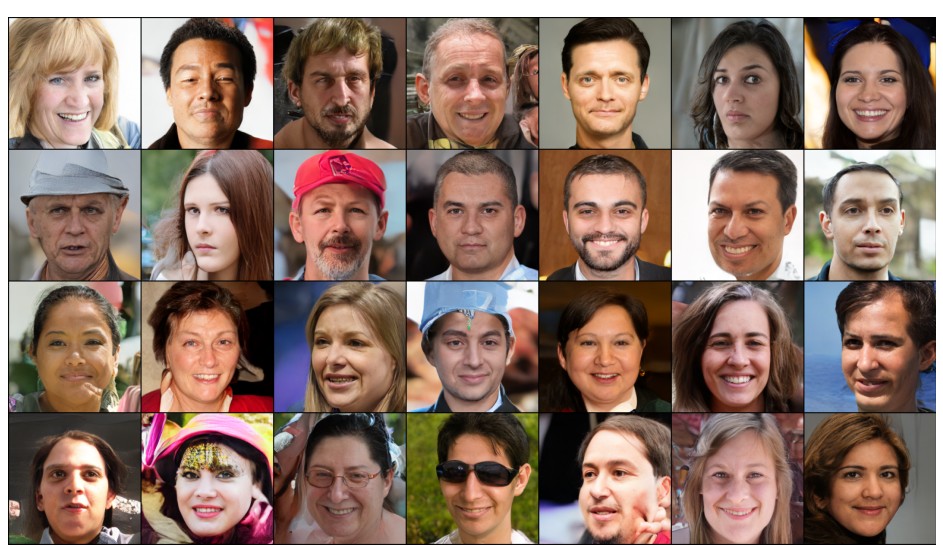

Figure 8: CATDM samples of unconditional image generation on FFHQ.

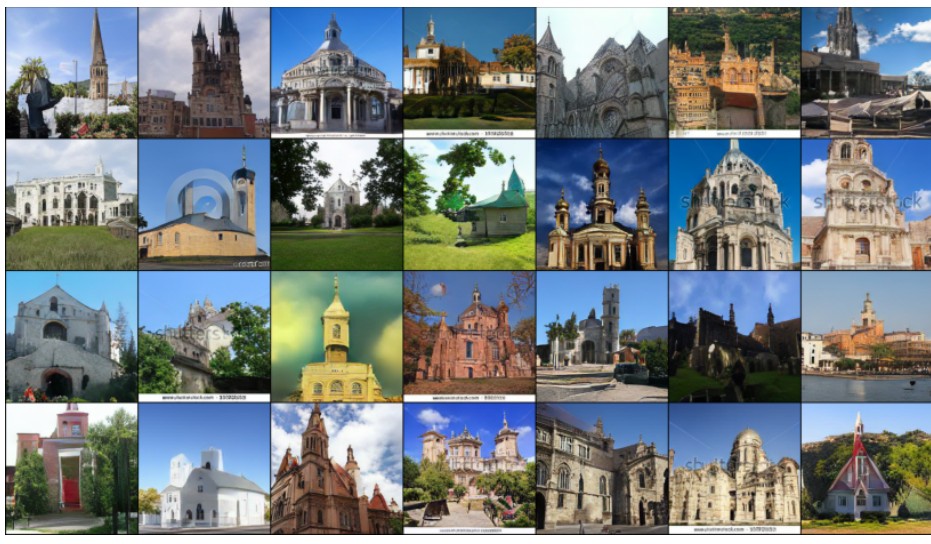

Figure 9: CATDM samples of unconditional image generation on LSUN Churches.

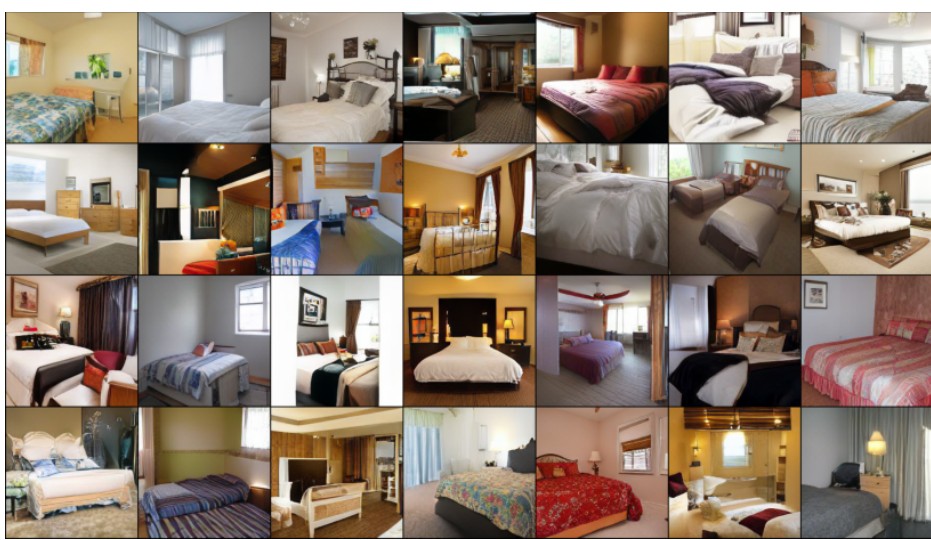

Figure 10: CATDM samples of unconditional image generation on LSUN Bedrooms.

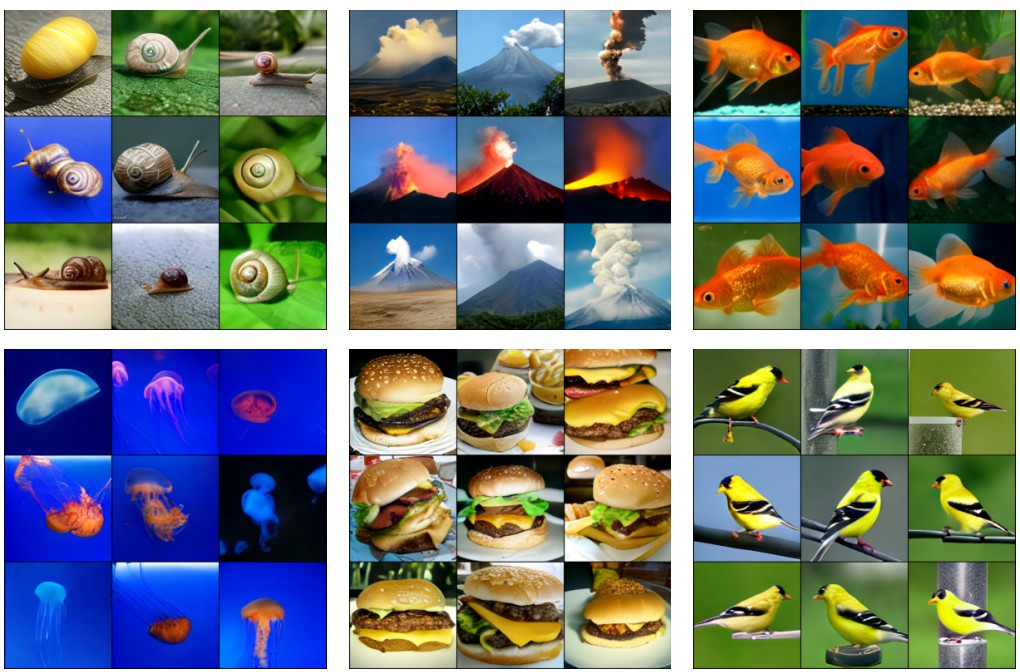

Figure 11: CATDM samples of conditional image generation on ImageNet $256 \times 256$ for selected classes, including "snail", "volcano", "goldfish", "jellyfish", "cheeseburger", "goldfinch".

