# OpenReview forum: "Mitigating Embedding Collapse in Diffusion Models for Categorical Data"
_ICLR.cc/2025/Conference — ICLR 2025 Conference Withdrawn Submission_

### Official Review · Reviewer_RWzq · 2024-11-03

**Soundness:** 2
**Presentation:** 2
**Contribution:** 2
**Rating:** 3
**Confidence:** 3

**Summary:**

The paper introduces CATDM, a continuous diffusion model designed to address the problem of embedding collapse when learning both embeddings and diffusion models for categorical data. CATDM incorporates a novel Consistency-Matching (CM) regularizer, a shifted cosine noise schedule, and a random dropping strategy to stabilize training and prevent collapse. The authors demonstrate that CATDM achieves superior performance across various benchmarks, including FFHQ, LSUN, and ImageNet for image generation, and shows competitive results in text generation and machine translation.

**Strengths:**

1)	The issue of embedding collapse is quite interesting and important in the context of diffusion models.
2)	The authors propose the use of a CM regularizer, which seems to alleviate this problem, although the approach itself does not appear to be particularly novel.

**Weaknesses:**

I have several concerns regarding the results presented in the paper:
1)	For image generation: Since your experiments are conducted in continuous space, it is unclear why comparisons with models like LDM/DiT and many other baselines were not included.
2)	For text generation: The BLEU scores for some of the baselines seem unusually low. You mentioned that all results, except for Difformer and CATDM, were taken from previous studies. However, for example, according to [1], SeqDiffuSeq's BLEU scores for WMT14 EN-DE and IWSLT14 DE-EN are significantly higher than the ones you reported, for WMT14 EN-DE, it is 19.76(ScareBLEU)/24.24(Tokenized BLEU) in [1] vs. 14.37 that you reported. Could you clarify the reason for this discrepancy?
3)	In your paper, you did not include comparisons with other relevant diffusion-based models. For example, [2] achieved 24.62 SacreBLEU on WMT14 EN-DE and 31.44 SacreBLEU on IWSLT14 DE-EN without using knowledge distillation (with Length Beam=10, MBR=5).
4)	Could you specify which type of BLEU score you are reporting—whether it is tokenized BLEU or SacreBLEU? Additionally, please clarify the decoding parameters used in your experiments (e.g., beam size, minimal Bayes risk, etc., if applicable). How were these parameters chosen for the baselines you compared against? Without these details, it is difficult to ensure that the comparisons are fair.
5)	Also it would be helpful if you could clearly indicate the source of the performance data for your model. If possible, please provide precise references (for example, indicating specific tables or sections in the paper). Currently, this information is missing, making it challenging to verify the claims made in the paper.
6)	Based on the current results, we find that the overall performance of CATDM appears relatively weak compared to other works, both in machine translation and Text8 benchmarks. Even if knowledge distillation was not used, it seems inaccurate to claim that CATDM achieves the best results among non-autoregressive models considering models like CMLM or many other models.
7)	The presentation of results for DE-EN and EN-DE in the figures and tables could be improved for clarity. Although they are labeled below the figures, it might still give the impression that these are two translation directions for the same dataset, which could confuse readers. We suggest reformatting this section to make it clearer.
8)	Several typos in the manuscript: around line 457, 478, and 933, respectively.

[1] SeqDiffuSeq: Text Diffusion with Encoder-Decoder Transformers. Hongyi Yuan, Zheng Yuan, Chuanqi Tan, Fei Huang, Songfang Huang, https://arxiv.org/abs/2212.10325
[2] DINOISER: Diffused Conditional Sequence Learning by Manipulating Noises. Jiasheng Ye, Zaixiang Zheng, Yu Bao, Lihua Qian, Mingxuan Wang, https://arxiv.org/abs/2302.10025

**Questions:**

Please refer to the weakness section.

---

### Official Review · Reviewer_GV8G · 2024-11-03

**Soundness:** 2
**Presentation:** 2
**Contribution:** 2
**Rating:** 3
**Confidence:** 4

**Summary:**

This paper handles in the situation of training reconstruction module and score matching module in the latent space together. This paper raise and analysis on the embedding collapse problem in the previous pre-trained embedding space. The new variance scheduling and consistency regularizer mitigate the embedding collapse.

**Strengths:**

- Since the paper targets on the categorical data, it can model both image and text modality.

- Applying consistency loss after softmax seems novel and argument in 4.3 seems reasonable.

**Weaknesses:**

1. The analysis on Figure 1 is not clear. For CSDM case, I know the variance of VQGAN embedding has small variance. This is often occurred when training auto-encoder without regularization. But, small variance does not indicates the meaningless embedding. The semantic meaning could be embedded in small region.

- 1.1. If analyze it on the pre-trained autoencoder with kl-regulaization, it must be have high variance. However, kl-regularization makes poor reconstruction. Why high variance on embedding space is important?

- 1.2. The author mentioned "cross-entropy loss for CSDM is low around t = 0 but increases rapidly as t arises". I can not catch why this is the problem. Even though this the problem, this can easily resolved by time scheduling (i.e. cosine). The visualization on the generative process maybe can give me the intuition.

2. Why CSDM in table 1 is so poor? I saw many latent diffusion model that has good FID performance. What is the difference in the FID of 12.66 in table 2?

3. The baseline in Table 3 are outdated. They are published at least 3 years ago. I saw many papers with FID lower than 2.0 recently.

**Questions:**

See weaknesses.

---

### Official Review · Reviewer_hpgr · 2024-11-03

**Soundness:** 2
**Presentation:** 1
**Contribution:** 2
**Rating:** 3
**Confidence:** 3

**Summary:**

This paper mainly deals with one problem of diffusion models for categorical data: embedding collapse. The authors provide reasons to explain why the embedding collapse happens in training. Concurrently, a novel method encompassing a new objective, the consistency-matching loss is proposed to solve the problem. The authors conduct experimental results to support the reasons and the method. They also provide useful theoretical support for their proposed method.

**Strengths:**

* The question of neural collapse in the embedding space is an intriguing question worth exploring, and the authors have provided an intuitive understanding of it with some usable experimental results as support.
* The authors have testified the algorithm in various settings against various baselines and have achieved comparatively good performances.
* The authors have provided supportive theoretical results toward optimality aspects of the algorithm although just in ideal cases.

**Weaknesses:**

1. **Performance Issues:** The experimental performances demonstrated in the experiments are generally not satisfactory enough and are outperformed by several already existing baselines.

2. **Motivation Behind Method Choices:**
   - Although the authors claim and show that the method has outperformed non-autoregressive methods they select as baselines, why should people further investigate improving non-autoregressive methods instead of switching to autoregressive ones for better performances is left unexplained.

3. **Writing and Structure Concerns:**
   - **Core Parts of the Writing:**
     - The method parts of the writing in this paper appear to be unpracticed and not acceptable for at least the following reasons:
       * Many of the introduced domain-specific concepts are improperly explained at the time they are introduced, including PFODE, variational diffusion formulation, FFHQ, etc. The authors should explicitly show how these concepts are related to the paper’s core topic and contributions at the time they are introduced.
       * Countless arguments of the paper can only be considered as highly personal insights or biased intuition, which are left without sufficient evidences or supportive references stated. For example:
         - In row 35: “The core concept is to progressively recover the original data distribution using a learned transition kernel.”
         - In row 247: “When the timesteps are small, the model learns the true categorical distribution through the reconstruction loss.”
         - In row 300: “Determining the right amount of noise added to the embeddings in each timestep can play an important role in both the forward and reverse processes of CATDM.”
         The authors should state out whether these are supported by literature or their experiments explicitly.
       * The paper should be better structured in order to deliver a clearer description of the method proposed, which is currently scattered into several sections and subsections left without a general overview to show the inner connections of these sections.

**Questions:**

* Could the authors provide explanations about why the method part of the paper is written in this structure? The current structure appears to be confusing and unorganized to me.
* Could the authors provide further information about the baseline choices? The baselines included in the paper are mostly dated before 2023 and are prone to be outdated according to my knowledge.

---

### Official Review · Reviewer_mcsW · 2024-11-03

**Soundness:** 2
**Presentation:** 3
**Contribution:** 2
**Rating:** 5
**Confidence:** 3

**Summary:**

This paper proposes a connection in continuous-space diffusion models between simultaneously optimizing  the reconstruction of discrete data embeddings and ensuring the consistency of cross entropy denoising on one hand, and optimizing the cross entropy denosing on the other.  The former two objectives are added to the usual diffusion model loss with the goal of preventing embedding collapse. To this end additional measures are taken, such as changing the noising schedule, re-weighting the three objectives in the loss and applying random dropping. Then the proposed model is applied to image generation, text generation, and machine translation.

**Strengths:**

The paper is clearly written and well organized, though notation indicating positions in the sequences could be improved. (clarity)

A re-weighting of the objectives in the loss function, combined with an improved noise schedule and a random dropping strategy improve results.(originality, significance)

The proposed model outperforms other tested methods on different data types and tasks. (quality)

**Weaknesses:**

1) Proposition 2 (and therefore the proof of Theorem 1) appears to be incorrect. More precisely, the conclusion in line 922 (appendix) is not true. If at $z_t$ (perturbed from $emb(x)$) the model predicts the one hot encoding for a particular token $x_i$ at position $i$ of the preperturbed sequence $x$, then this prediction will be incorrect for all other sequences $y$ that generate $z_t$, and that coincide with x in all but position $i$. The true optimal model is the ground truth $p_{0|t}(x^i|z_t)$. That is, loss $L_{CE}=-E_{p(x)}E_{p(z_t|x)}\sum_{i=1}^M \log{p^{\theta}_{0|t}(x^0_i=x_i|z_t)}$ is minimized when

$p^\theta_{0|t}(x_i|z_t)=p_{0|t}(x_i|z_t)$. In this case $L_{CE} = \sum_{i=1}^M E_{p(z_t)}H(p^{\theta}_{0|t}(x_i|z_t))>0.$ I would be grateful if the authors could discuss this.

2) Assuming the correctness of proposition 2: The paper proposes only that optimizing the two objectives ($ L_{CM} $ and $L_0$,) optimizes the latter $L_{CE}$ . However, using the same arguments, the other direction should hold as well, as indeed having a perfect denoising cross entropy loss, the same arguments imply perfect reconstruction at time $t=0$, and since the model predicts the correct unperturbed token (for each position) at each time point, then such predictions will be constant (consistent). As such it is difficult to see why one should use these two more convoluted objectives instead of just using denoising cross entropy which itself mitigates embedding collapse. In addition, then $\mathcal{L}_{DM}$ is added which theoretically is not necessary (according to Theorem 1), and in practice introduces the issue of embedding collapse with its $L_2$ loss. So my question is why not simply use the denoising cross entropy alone as in Dieleman et al. 2022? It is essential to compare the proposed method against the pure denosing cross entropy loss on the same task using the same model.

3) The paper applies the method mostly to inherently continuous data (which is quantized), while the title and premise of the paper imply improving the modeling of categorical data. It would have been better to focus more on text or protein design. However, the experiments on text are very limited, testing only on text8 at character level, where they underperform in comparison to categorical diffusion models. On machine translation the results are better but the paper does not provide results for SEDD of Lou et al 2023 in this case.

4) Benchmark models for continuous data are not recent. Most recent models have FIDs under 2, but the proposed model shows a FID of 6.81.

**Minor Issues**

In line 108, shouldn't it be *backward* process?

Line 124: it is a bit confusing stating element $i$, as this normally means dimension $i$ of a vector. However in this context we are referring to the embedding at position $i$ in the sequence embedding.

Expression in 240, does not really coincide with consistency models of Song et al 2023. Their models are discrete in time, the continuous version has a different form (see Remark 10 in the appendix of their paper). In their loss only times $t_n$ and $t_{n+1}$ appear, and not non-neighbor times.

**Questions:**

How does the proposed model  compare against SEDD (Lou et al 2022) and Dieleman et al 2022 on Openwebtext (or a subset) utilizing the tokenizer of GPT2, on short length sequences of 128 (in all cases utilizing the network of Lou et al 2023)?

What is the training speed when utilizing $ L_{CM} $? How is it implemented in practice? A detailed algorithm would be very helpful.

Assuming the correctness of proposition 2: My main question is why use $ L_{CM} $ and $L_0$, instead of  $L_{CE}$ alone? Then why also add  $L_{DM}$? I understand that the paper starts from diffusion models, but once $ L_{CM} $ and $L_0$ are added, theoretically (as it is shown in Theorem 1) they should be enough to learn the paths without adding $L_{DM}$. I would be grateful if authors could add another row in Table 1, where only $ L_{CM} $ + $L_0$ is used.

---

### Note · Authors · 2024-11-25

**Comment:**

We sincerely thank the reviewers for their time and effort in evaluating our manuscript. After carefully reflecting on your valuable feedback, we have concluded that the  manuscript would benefit from more comprehensive experiments and improving clarity. Therefore, we have decided to withdraw our submission to further improve the work.

**Withdrawal Confirmation:**

I have read and agree with the venue's withdrawal policy on behalf of myself and my co-authors.